# Spatial genomic heterogeneity in multiple myeloma revealed by multi-region sequencing

L. Rasche[1], S.S. Chavan[1], O.W. Stephens[1], P.H. Patel[1], R. Tytarenko[1], C. Ashby[1], M. Bauer [1], C. Stein[1], S. Deshpande[1], C. Wardell[1], T. Buzder[1], G. Molnar[1], M. Zangari[1], F. van Rhee[1], S. Thanendrarajan[1], C. Schinke[1], J. Epstein[1], F.E. Davies[1], B.A. Walker [1], T. Meissner[2], B. Barlogie[1], G.J. Morgan[1] & N. Weinhold[1]

In multiple myeloma malignant plasma cells expand within the bone marrow. Since this site is well-perfused, a rapid dissemination of "fitter" clones may be anticipated. However, an imbalanced distribution of multiple myeloma is frequently observed in medical imaging. Here, we perform multi-region sequencing, including iliac crest and radiology-guided focal lesion specimens from 51 patients to gain insight into the spatial clonal architecture. We demonstrate spatial genomic heterogeneity in more than 75% of patients, including inactivation of CDKN2C and TP53, and mutations affecting mitogen-activated protein kinase genes. We show that the extent of spatial heterogeneity is positively associated with the size of biopsied focal lesions consistent with regional outgrowth of advanced clones. The results support a model for multiple myeloma progression with clonal sweeps in the early phase and regional evolution in advanced disease. We suggest that multi-region investigations are critical to understanding intra-patient heterogeneity and the evolutionary processes in multiple myeloma.

[1] Myeloma Institute, University of Arkansas for Medical Sciences, Little Rock, AR 72205, USA. [2] Department of Molecular and Experimental Medicine, Avera Cancer Institute, Sioux Falls, SD 57105, USA. Correspondence and requests for materials should be addressed to N.W. (email: NWeinhold@uams.edu)

Multiple myeloma (MM) is the malignant counterpart of an antibody-secreting terminally differentiated B cell—a plasma cell (PC). MM is not a single disease but is rather comprised of a number of molecular subgroups characterized by specific chromosomal aberrations[1]. Acquired genomic events associated with progression lead to further inter- and intra-patient clonal heterogeneity, translating into different clinical outcomes[2, 3]. MM is a particularly interesting model system to gain insights into the molecular progression in cancer because its mutational load falls in the middle of the range of mutations seen in cancer lying somewhere between a genetically complex solid malignancy constrained in its evolution by neighboring cells and the genetically simple leukemia, which is distributed throughout the bone marrow (BM) with less anatomical constraints[2].

Molecular data available to date have been interpreted in the context of a disease model where MM is the end result of a multistep transformation process during which the acquisition of genetic hits leads to the generation of branching evolutionary pathways[2, 4]. According to this Darwinian model, the acquisition of sequential mutations results in better adaptation of clonal cells to their microenvironment leading to the outgrowth of "fitter" clones, which outcompete previously dominant clones[2]. Similar to leukemia, MM primarily grows in the BM, where free movement between sites through the circulation is assumed, a feature consistent with small numbers of clonal cells being seen in the peripheral blood using flow cytometry[5, 6]. Based on this observation and in contrast to solid cancers[7, 8], a rapid and homogenous dissemination of "fitter" clones throughout the BM-containing skeletal system may be anticipated. However, for MM this assumption is not proven, and in fact whole-body imaging with positron emission tomography (PET) is consistent with an imbalanced distribution of the disease[9]. This is highlighted by the fact that up to 80% of newly diagnosed patients present with focal accumulations of malignant PCs in restricted areas within the BM; so called focal lesions (FL), that are often superimposed on diffuse interstitial growth patterns[10]. The number of FLs (on average 18 per patient) is prognostic indicating a significant contribution to tumor progression[9–12]. Moreover, recent longitudinal analyses suggest branching evolution pathways and thus the pre-existence of clones that drive relapse[13–15]. Failure to detect such clones at baseline could be due to insufficient test sensitivity but it could also be explained by regionally restricted evolution. Since the diagnosis of MM is usually based on "blind" BM sampling from a single site, predominantly at the posterior iliac crest, such local evolution would not be apparent.

To avoid this sampling bias our diagnostic approach includes examinations of interventional radiology-guided aspirates from FLs along with traditional iliac crest biopsies. Here we present the genomic analysis of a set of paired samples derived as part of our clinical routine program (Fig. 1, Supplementary Table 1 and Supplementary Data 1). By multi-region sequencing on 42 newly diagnosed and 11 treated MM patients, we demonstrate spatial genomic heterogeneity in the majority of patients. Furthermore, we show that the extent of spatial heterogeneity is positively associated with the size of biopsied FLs consistent with regional outgrowth of advanced clones. We suggest that multi-region investigations are critical to understanding intra-patient heterogeneity and the evolutionary processes in MM.

## Results

**Spatial genomic heterogeneity in newly diagnosed MM.** Genomic heterogeneity can be deciphered through the analysis of numerical or structural chromosomal aberrations, short insertions or deletions (Indels) or single nucleotide variants (SNVs), and we describe heterogeneity at all of these levels.

**Heterogeneity at the chromosomal level.** We used high-resolution single nucleotide polymorphism (SNP) arrays and WES data to call copy number aberrations (CNAs). To account for the sensitivity of this approach for detection of subclones (threshold: ~20%) we only included CNAs that were clonal in at least one of the paired samples. Using this strategy we found spatial differences in chromosomal profiles in 17 of the 42 (40%) newly diagnosed patients with, on average, three unshared CNAs (range 1–28, Fig. 2a, Supplementary Fig. 1A and Supplementary Data 2). Investigating commonly used prognostic markers, we found examples of spatial differences for all of them; the key adverse prognostic marker, del(17p), showed spatial variation in two of six patients (33%), followed by translocations involving the MYC locus (t(MYC)), which was a site-specific event in four of sixteen patients (25%) who carried this translocation (Fig. 2a). Although del(1p) and gain/amplification of 1q21, important changes associated with progression[16, 17], were frequently shared between different spatial sites, four of twenty-one patients (19%) presented with regionally restricted events (Fig. 2a). Interestingly, loss of heterozygosity (LOH), such as LOH of 1q, often contributed to spatial heterogeneity, and was a non-ubiquitous event in nine patients (21%, Supplementary Data 2). Changes on chromosome 1 and 4 were the most frequent contributors to spatial heterogeneity ($n = 7$ patients), followed by chromosome 5 and 8 ($n = 6$) including deletions, gains and LOH.

Importantly, the primary etiological IgH translocations, such as t(4;14) and t(11;14), were consistently shared between regions, in line with their role as initiating events[1]. However, for two hyperdiploid (HRD) cases, the other potentially etiologic subgroup, discordance between sites was seen (Supplementary Fig. 2). In one case with a HRD karyotype at the iliac crest, several large-scale deletions were present in an FL at the fourth lumbar vertebra, which gave rise to a total number of 46 chromosomes, formally corresponding to a non-HRD karyotype. In the second patient, we found a non-HRD karyotype at the right posterior iliac crest and an atypical HRD karyotype with trisomies of the even-numbered chromosomes 2, 4 16, 20 and 22, in a nearby FL located in the pelvis.

**Heterogeneity at the mutational level.** We performed a combined analysis of non-silent SNVs and Indels in which mutations not ubiquitously detectable were classified as "unshared" (please see methods for the detection threshold and further details). Ubiquitous mutations, with at least a three-fold difference in cancer clonal fraction (CCF) between paired samples were termed shared-differential ("shared-diff"). We also regarded these shared-diff variants as being heterogeneous mutations as they are consistent with a considerably different clonal structure between sites.

The analysis of mutational profiles showed that genomic heterogeneity in space was more pronounced than was seen at the CNA level (Fig. 2c). Unshared and shared-diff mutations were found in 32 (76%) and 30 (71%) of newly diagnosed patients, respectively (Supplementary Data 3) and all patients with CNA differences also showed heterogeneity at the mutation level. Of note, the proportion of heterogeneous mutations varied considerably between patients with three patients showing >50% unshared mutations and another three patients presenting without any detectable genomic heterogeneity. The total number of non-silent mutations (median: 50, range: 19–435, Fig. 2b) per patient did not significantly correlate with the proportion of unshared and shared-diff mutations (Spearman's rank correlation $\rho = 0.3$ ($P = 0.055$) and $\rho = 0$ ($P = 1$), respectively).

We performed a more detailed analysis of the most frequently mutated genes (NRAS, TTN, KRAS, ROBO2, BRAF, PCLO,

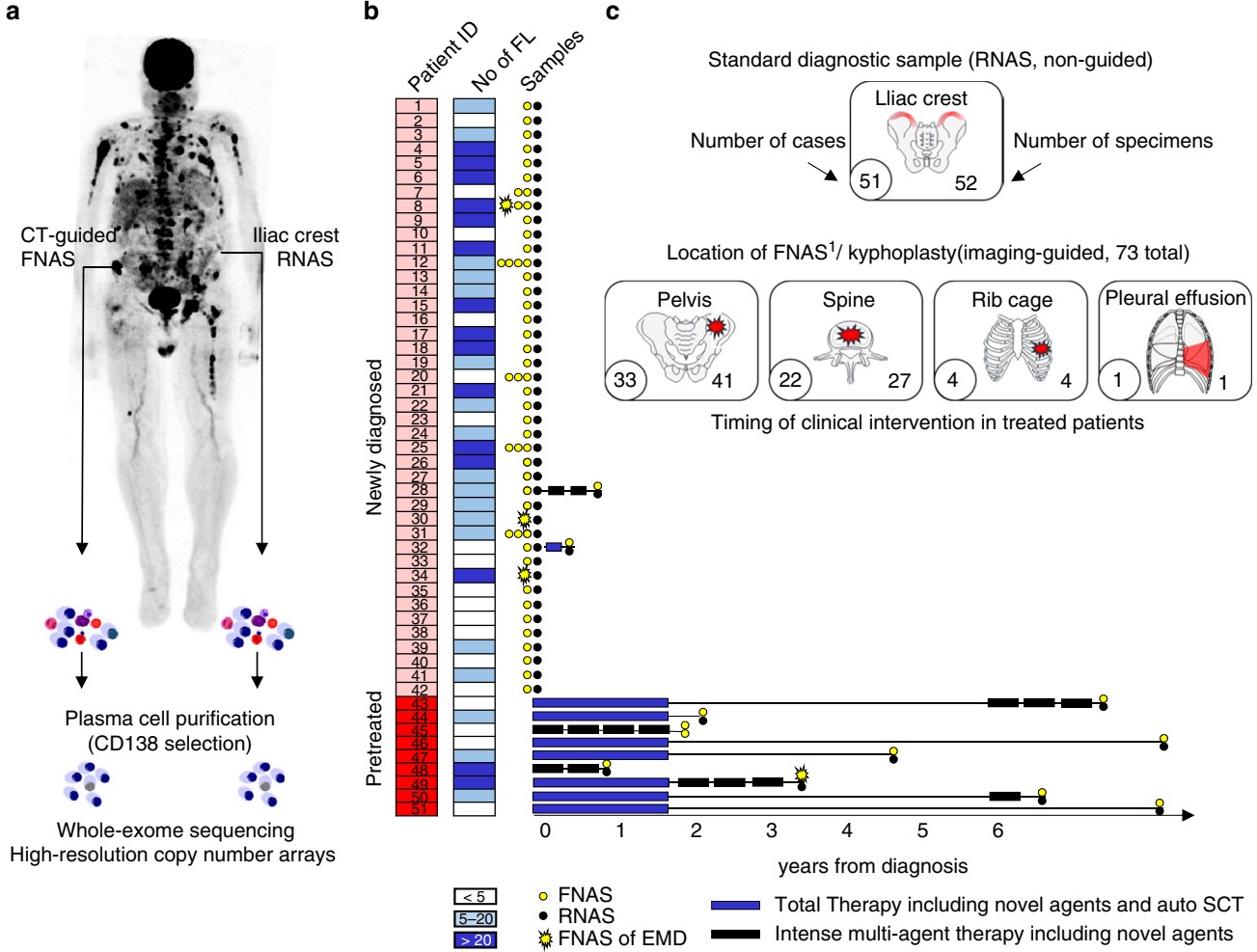

**Fig. 1** Sample origin and processing. **a** Traditional iliac crest specimens (random aspirates) and CT-guided fine needle aspirates were enriched for malignant plasma cells using CD138-positive selection. Mutational and chromosomal profiles were investigated using whole-exome sequencing and high-resolution copy number arrays. **b** Disease state at sampling, treatment history, the total number of focal lesions in medical imaging, and the number and type of investigated samples are depicted. **c** Origin and number of samples. *FL* focal lesion, *FNAS* fine needle aspirate, *RNAS* random aspirate, *EMD* extramedullary disease, *auto SCT* autologous stem cell transplantation

*TRAF3, TET2, FAM46C,* and *DIS3*) and other recurrently mutated genes in MM[1]. The top five genes showing unshared and/or shared-diff mutations in this set were *NRAS, KRAS, TTN, ROBO2,* and *BRAF* (*dark blue* and *green squares* in Fig. 2d). We note that alterations of genes involved in the mitogen-activated protein kinase (MAPK) pathway were by far the most frequent events contributing to heterogeneity in space at the mutation level. Importantly, we found patients with unshared ultra-high-risk bi-allelic events affecting the tumor suppressor genes *CDKN2C* (*n* = 1) and *TP53* (*n* = 2)[13, 18].

Extending this analysis to include all genes with spatial heterogeneity demonstrated that the most frequently mutated genes were not necessarily the ones with the greatest frequency of unshared or shared-diff events. In this respect *ANK1* and *MTR*, two genes mutated in three patients, all showed spatial heterogeneity (Supplementary Fig. 3). Conversely, not all frequently mutated genes showed heterogeneity. *ATM* (*n* = 3), *PCLO* (*n* = 6), *TET2* (*n* = 4), and *TRAF3* (*n* = 4) all had shared mutations occurring at similar frequencies (Fig. 2d). However, even mutations in these genes were frequently subclonal.

**Heterogeneity in space despite treatment.** The therapeutic aim of multi-agent chemotherapy, such as the one used in the Total

Therapy (TT) protocols[19], is to increase the depth of response and to eradicate residual resistant clones. Thus, for initially responding patients we would expect relapse to be dominated by a limited number of highly resistant selected clones, and as a result less spatial heterogeneity. To test this hypothesis we analyzed a set of heavily treated patients at different stages of MM therapy (Fig. 1 and Supplementary Data 4). All patients presented with adverse prognostic markers on chromosome 1 and all but two had a mutation in genes of the MAPK pathway, indicating a strong selective pressure of treatment (Fig. 2d). However, we still observed ongoing events at 1q leading to a further amplification of this region at specific sites in two patients (5 vs. 4 and 4 vs. 3 copies, respectively), regionally restricted bi-allelic inactivation of *RB1*, as well as a case with two spatially separated dominant clones defined by *NRAS* and *KRAS* mutations (Figs. 2a, d). The average level of heterogeneity on the chromosomal and the mutational level did not significantly deviate from the corresponding values in newly diagnosed patients (Wilcoxon tests, *P* > 0.05). Together, these results indicate a strong selective pressure of treatment and further regional evolution of selected clones.

**Site-specific high-risk clones drive prognosis.** Risk stratification, including fluorescence in situ hybridization and gene expression

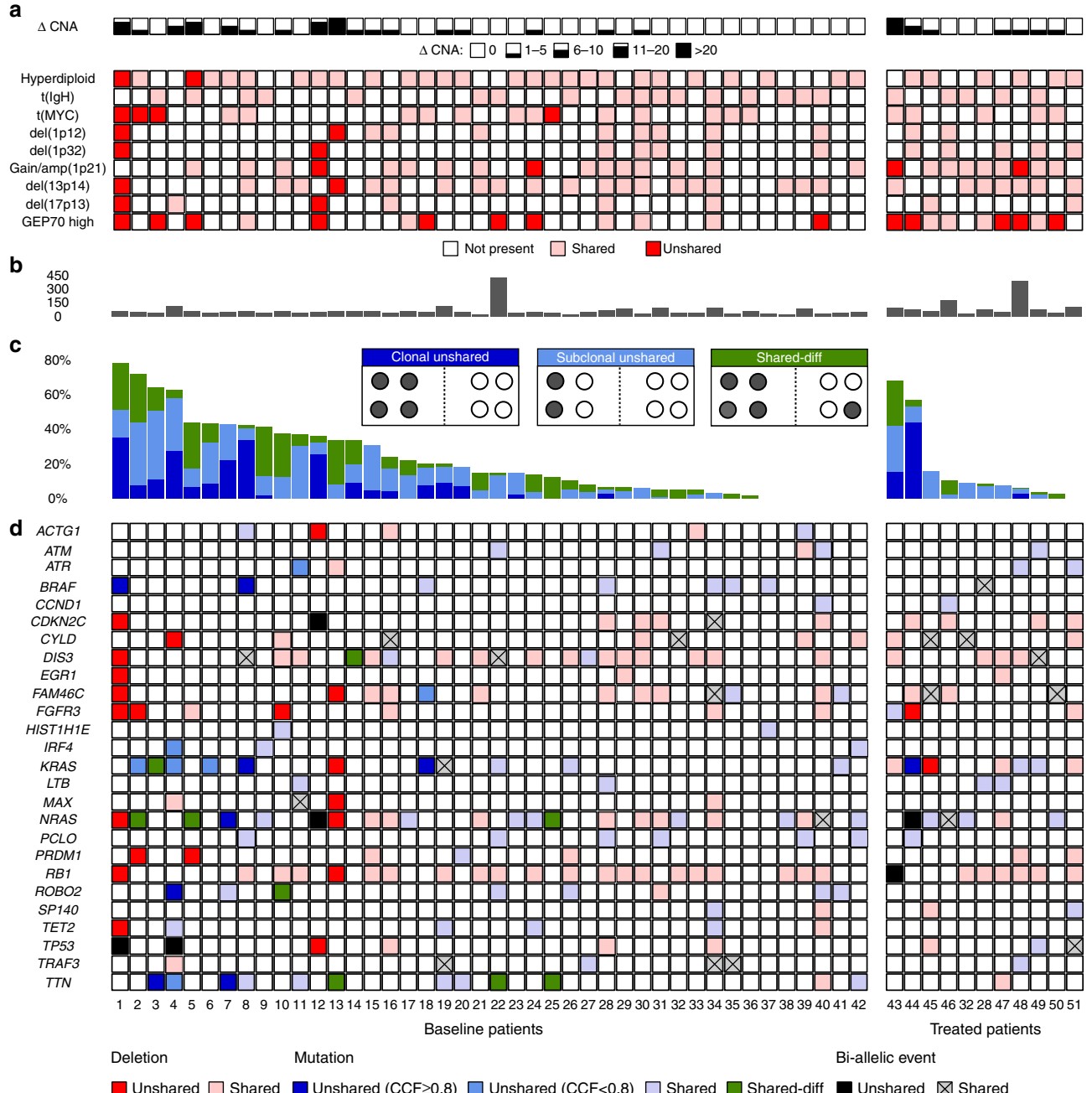

**Fig. 2** Spatial heterogeneity in baseline and treated patients. In the upper panel of **a** the number of unshared copy number aberrations between paired iliac crest and fine needle aspirates (Δ CNA, > 1 Mb) is shown. In the lower panel of **a** heterogeneity involving the initiating events hyperdiploidy and recurrent IgH translocations as well as commonly used prognostic markers is presented with *light and dark red* indicating shared and unshared events, respectively. The total number of non-silent mutations is presented in **b**. **c** Proportion of clonal unshared (cancer clonal fraction (CCF) ≥ 0.8; *dark blue*), subclonal unshared (CCF < 0.8; *cornflower blue*) and shared-diff (*green*) non-silent mutations. In **d**, non-silent mutations and deletions affecting the most frequently mutated genes in our set and other recurrently mutated genes in multiple myeloma are shown. *Dark blue* and *cornflower blue* denote clonal and subclonal *unshared* mutations, *green* shared-diff mutations, *red* deletions and *black* bi-allelic events. *Light colors* or *crossed boxes* indicate shared events

profiling (GEP) data has become a component of routine diagnostics in MM, and can be used to direct treatment[20–23]. In the set of patients analyzed 25% (13 cases) showed discrepancies in the GEP70 risk status[24] between different sites, e.g. patient no. 1 (Fig. 3a). In this case the dominant GEP70 low-risk clone at the iliac crest had a HRD karyotype and also contained a site-specific t(MYC) and an actionable *BRAF* V600E mutation. Importantly, and in contrast to the iliac crest clone, the dominant clone at L4 was GEP70 high-risk associated with an ultra–high–risk bi-allelic *TP53* deletion and an adverse event involving chromosome 1p.

The absence of the *BRAF* mutation at L4 was confirmed by targeted sequencing (654×). In this case a diagnostic approach based on solely assessing an iliac crest specimen would falsely assign this patient to the GEP70 low-risk group, and if BRAF V600E was used for a targeted treatment approach it would fail at the GEP70 high-risk site.

In order to understand the clinical implications of this concept and to elucidate the prognostic impact of site-specific high-risk clones, we analyzed an extended set of 263 patients enrolled in TT protocols (Supplementary Table 1). These cases had GEP70 risk

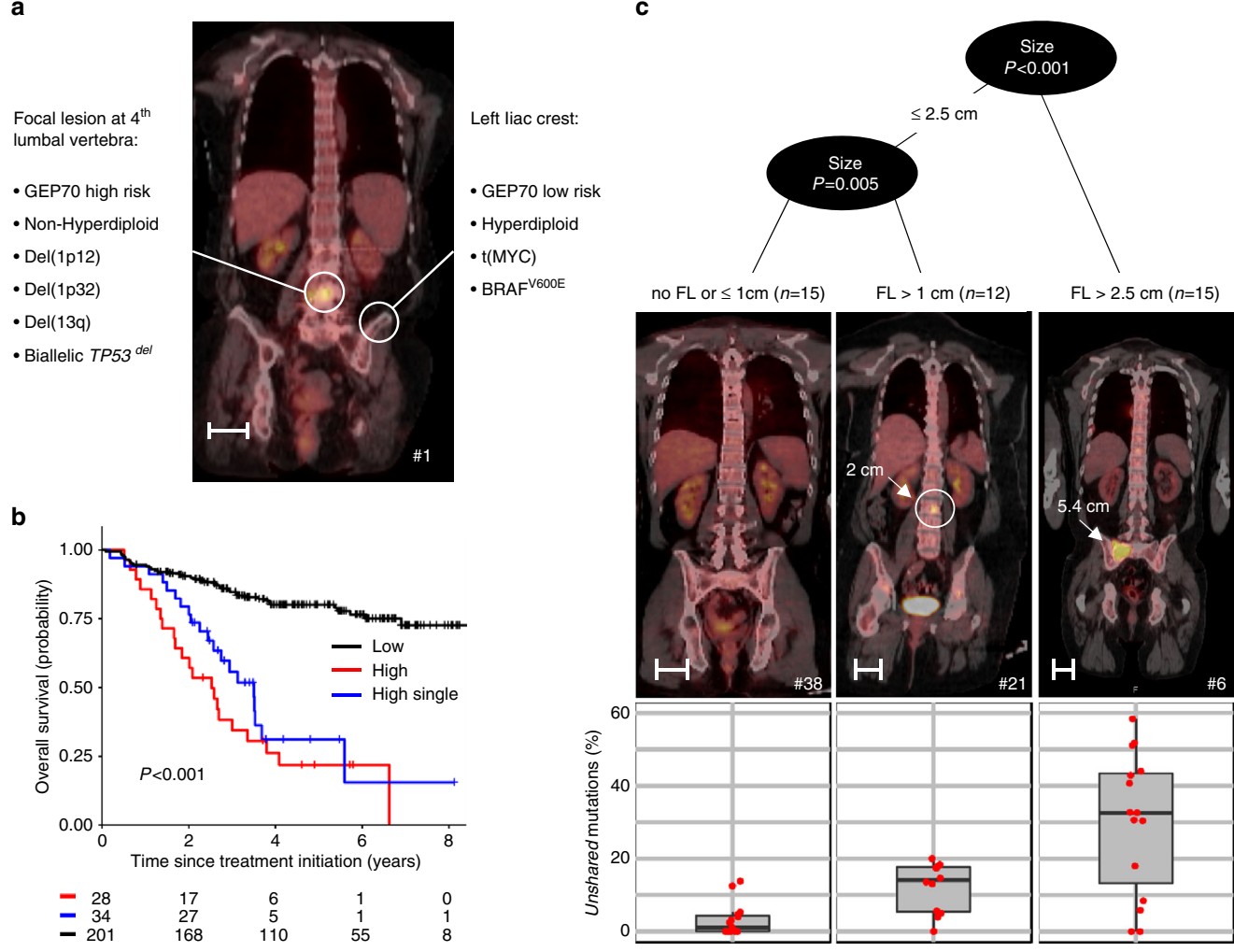

**Fig. 3** Spatial heterogeneity: example, impact on outcome and association with the size of focal lesions. In **a**, unshared key drivers and the risk status in paired samples of patient no. 1 are shown. **b** Overall survival after enrollment into Total Therapy for 263 patients stratified by the GEP70 risk status in a routinely collected iliac crest sample and a paired CT-guided fine needle aspirate. Whereas *black* and *red* indicate patients with GEP70 low- and high-risk in both samples, respectively, the *blue line* shows the outcome for patients with discrepancies for paired samples (single-site high-risk). In **c**, recursive partitioning with the maximum size of focal lesions as the predictor of the proportion of unshared mutations was performed. Representative PET-CT images for the three size-groups are depicted (length of *scale bars* = 5 cm)

scores for paired iliac crest samples and computer tomography (CT)-guided focal lesion aspirates. This series had a median follow-up of 4.8 years with 13% (34 patients, 20 with high risk according to the FL sample only) having discordant GEP70 scores. The survival analysis showed a poor outcome for cases with a non-homogenous distribution of GEP70 high-risk clones which was similar to the outcomes for cases with GEP70 high-risk at both sites (28 cases) (Cox model, $P = 0.2$, Fig. 3b). This result suggests that high-risk subclones drive prognosis even if they are not ubiquitously distributed.

**Size of FL and regional dominance.** We investigated whether spatial heterogeneity could be predicted using other disease features. The analysis, after correction for multiple testing, showed that none of the standard clinical and molecular variables including the International Staging System (ISS)[25], ploidy, high-risk cytogenetics, and the GEP70 risk status at the iliac crest site were significantly associated with spatial heterogeneity in newly diagnosed patients (linear regression and Wilcoxon tests, respectively; $P > 0.05$, Supplementary Figs. 4–7). However, a trend was seen for a lower proportion of shared-diff mutations at

higher ISS stages and in cases with deletions involving 1p at the iliac crest site (Supplementary Figs. 4a and 6c). When whole-body imaging data was included into the analysis we identified a highly significant positive correlation between the maximum size of the biopsied FLs with the proportion of unshared mutations (Spearman's correlation $\rho = 0.62$, $P < 0.001$). Recursive partitioning utilizing the maximum size of FLs as the predictor of heterogeneity gave three groups with increasing levels of unshared mutations with cutoffs at 1 cm and 2.5 cm diameter, respectively (Fig. 3c). In contrast, neither the total number of FLs nor the anatomical distance between investigated sites correlated with spatial heterogeneity (Supplementary Fig. 8). Furthermore, spatial heterogeneity characterized by a different subclonal composition (shared-diff type) could not be predicted by any of the imaging-based parameters ($P > 0.05$).

**Multi-regional evolutionary events underlie disease progression.** To address whether the genomic profile of a single FL is representative of other FLs in the same patient, we investigated the phylogenetic relationship between clones at different sites in four patients with large FLs (>2.5 cm). In order to illustrate the

"main" events and branches, we focused on clonal aberrations. The results of this analysis clearly demonstrated the complexity of the evolutionary processes underlying the development of MM (Fig. 4). In patient no. 12, four FLs, which were classified as GEP70 high-risk, showed very similar genomic profiles, sharing the high-risk chromosomal events bi-allelic deletion of *CDKN2C*, gain(1q), and del(17p13) (Fig. 4a). These events were not found in the GEP70 low-risk clone, which was seen at a randomly biopsied non-FL site. The results of this analysis indicate that these FLs have a common high-risk ancestor which disseminates in a metastatic fashion on a background of GEP70 low-risk disease.

In contrast to this case, the FLs seen in patient no. 8 displayed different genomic profiles with each of them containing unique driver mutations (Fig. 4b). An FL at the left chest wall was characterized by a unique *KRAS* mutation and LOH of 1q. 1q was amplified in all FLs, but only this one showed LOH. In contrast an FL present at L1 had a unique *BRAF* mutation and a mutated *STAT3*, an event that was also found in a third FL. However, the site of the mutation differed between the two FLs (STAT3$^{Asn553Lys}$ vs. STAT3$^{Asp663Tyr}$). All sites investigated in this patient shared a t(MYC) and an amplification of 1q, supporting a common advanced ancestor, which was further changed during its multi-focal progression. This evolutionary pattern, driven by site-specific mutations, represents a route to disease progression, which is distinct from that seen in patient no. 12.

In a third case (no. 7) two evolutionary branches were observed (Fig. 4c). Tumor cells at T8 and the right iliac crest shared a missense *NRAS* mutation that was not detectable at the left ilium. In contrast to the *NRAS* branch, the large left ilium FL (6 × 2 × 2 cm) showed multiple site-specific CNAs including deletions affecting trisomic chromosomes as well as numerous unique non-silent mutations. This case illustrates both the impact of FL size and the lack of correlation between the anatomical distance between investigated sites and the presence of spatial heterogeneity.

A further type of progression was seen in case no. 20 where two major subclones occupied four distant sites (Fig. 4d): the first clone, characterized by an *IL6ST* mutation, was detected at the acetabulum and at the twelfth thoracic vertebra (T12), the second clone infiltrated the left iliac crest and the eighth thoracic vertebra (T8). When taking subclonal aberrations (CCF < 0.8) into account, we found an independent minor subclone with a *KRAS* mutation that infiltrated both sites in the pelvis and T8, but not T12.

**Non-neutral evolution in MM.** The evolution patterns described above strongly indicate non-neutral evolution. Recently, a mathematical model to discriminate clonal selection from neutral evolution was introduced[26]. This model utilizes

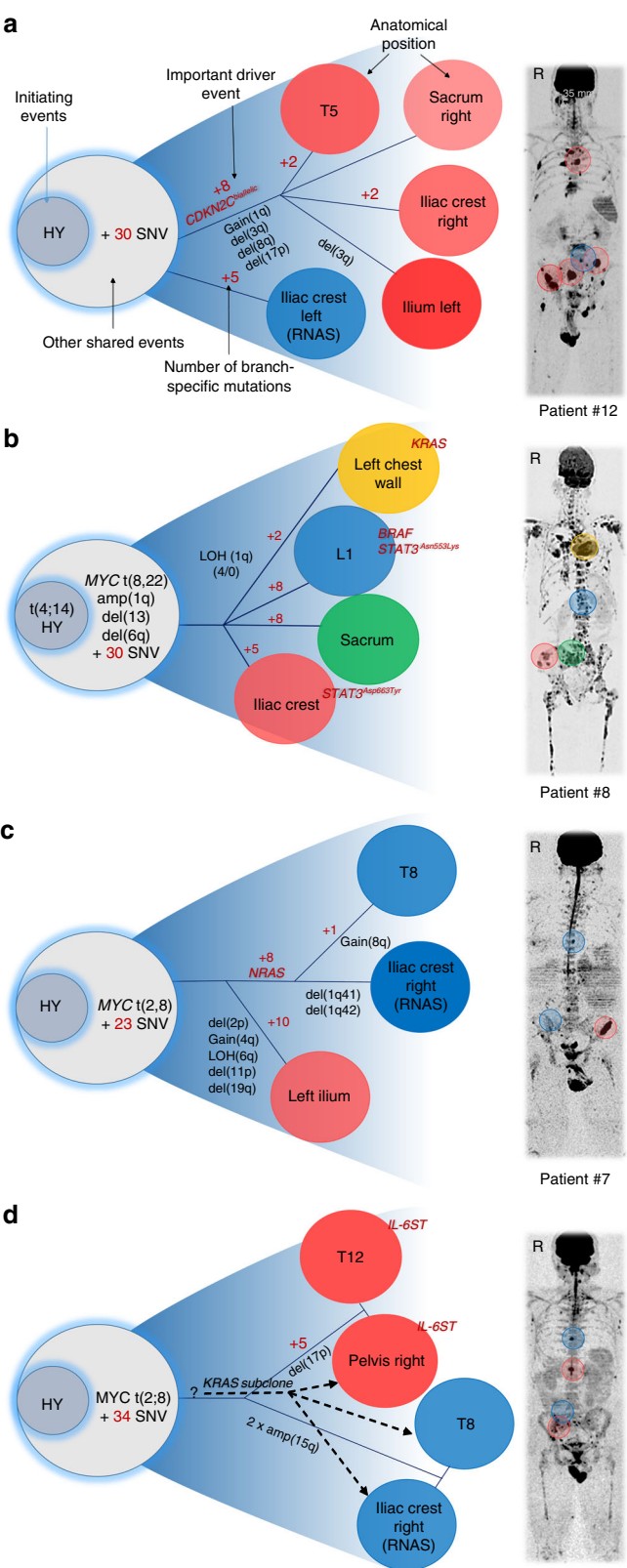

**Fig. 4** Multi-regional evolution. Cases with availability of multiple CT-guided samples were selected to analyze the phylogenetic relationship of clones from different regions. The location of samples is marked in the medical images in the *right panel* using the color code that was assigned to clones (*left panel*). The letter *R* indicates the right side of the body. **a** Four focal lesions (*FL*) showed similar genomic profiles, sharing the high-risk events bi-allelic deletion of *CDKN2C*, gain(1q) and del(17p). In contrast, a randomly biopsied non-FL site was low-risk and showed none of these events. **b** FLs displayed different genomic profiles with each of them containing unique driver mutations (*BRAF*, *KRAS*, and *STAT3*). Clones at the iliac crest and L1 contained different *STAT3* mutations (STAT3$^{Asn553Lys}$ vs. STAT3$^{Asp663Tyr}$). **c** Clones at T8 and the left iliac crest shared a missense *NRAS* mutation. Compared to the *NRAS* branch, the right iliac crest FL showed multiple site-specific CNAs and non-silent mutations. **d** Two major clones occupied four distant sites with the first clone being characterized by a non-ubiquitous *IL6ST* mutation. Moreover, an independent minor subclone with a missense *KRAS* mutation infiltrated both sites in the pelvis and T8, but not T12

subclonal aberrations with a variant allele frequency (VAF) boundary of (0.12,0.24) to provide information on growth dynamics[26]. We applied this model to mutation data from newly diagnosed patients to determine the predominant type of evolution in MM. In total, 62 of 99 samples were not included as fewer than 12 mutations were within the specified VAF boundary (Supplementary Data 5). None of the remaining 37 samples presented with a VAF distribution characteristic of neutral growth ($R^2 \geq 0.98$, two examples are shown in Supplementary Fig. 9). Further clonality analyses revealed that in 25 patients with heterogeneous mutations different clones dominated at different sites, further indicating selection instead of neutral evolutionary growth patterns. The clonal substructure in patient no. 1 is shown as an example in Supplementary Fig. 10.

## Discussion

This multi-region genomic analysis is the first systematic study that describes spatial heterogeneity in MM; a phenomenon that has been recently shown for several solid cancers[8, 26–31]. Furthermore, our observations are supported by two case reports describing spatial heterogeneity in one MM patient with extra-medullary disease[32] and in one relapse-refractory patient treated with the BRAF inhibitor Vemurafenib[33]. So far, MM has been assumed to be a cancer characterized by dissemination of "fitter" clones replacing prior, less fit subclones, by Darwinian selective sweeps[2]. In this regard, spatial heterogeneity is a perplexing observation, challenging this current model.

Recently, alternative evolution models have been introduced in the field of solid cancers providing a new way to interpret heterogeneity in space. Accordingly, cancer growth can be neutral with the distribution of mutations being determined by the time point at which they arise rather than by the fitness advantage they provide. Using a mathematical model to predict neutral evolutionary dynamics, one third of cancers showed this type of evolution[7, 26, 28]. However, the results of this current study allow us to assign MM to the class of malignancies defined by non-neutral evolution characterized by ongoing clonal competition and adaption to the microenvironment. The evidence supporting non-neutral evolution includes: (1) Site-specific driver aberrations were a frequent observation, which are not expected in neutral evolutionary systems where key driver mutations are usually shared events[30], (2) none of the tumors investigated followed the power-law distribution characteristic of neutral evolutionary growth patterns, (3) heterogeneous mutations frequently had high subclonal percentages ($CCF \geq 0.8$), supporting a model where clones are enriched at different sites, and (4) the extent of unshared mutations and CNAs correlated positively with the size of FL consistent with them driving progression at those sites.

Thus, clonal competition is the most likely explanation for increased spatial heterogeneity in patients presenting with FLs. A key feature in PC biology is competition for limited BM survival niches between long-lived PCs and new plasmablasts, resulting in the apoptosis of replaced cells[34]. This dependence on the BM microenvironment generally also holds true for malignant PCs, which usually do not survive in vitro[35]. In a situation where niches at different sites are already occupied by advanced clones, the fitness level of competing clones would need to be much more advanced to enable invasion and replacement. Thus regionally restricted evolution would be anticipated at later stages of disease where almost all niches are occupied, rather than clonal sweeps replacing prior subclones. This phenomenon is highlighted by the description of a case with four different dominant clones at four different sites. We appreciate that, in analogy to the neutral

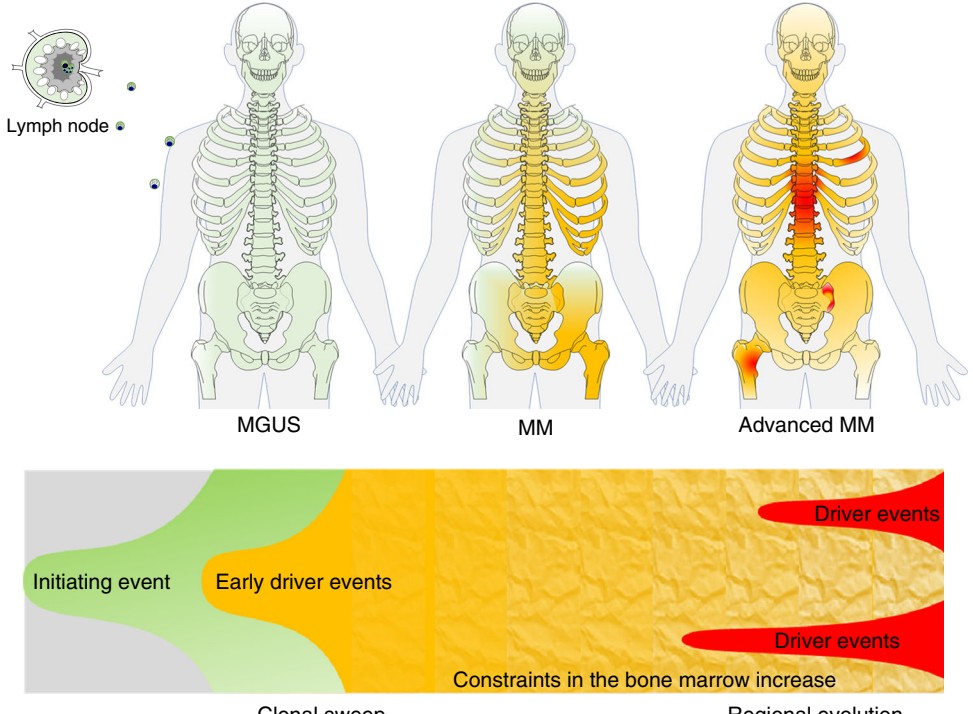

**Fig. 5** Regional differences in the context of a non-neutral evolutionary model considering spatial constraints. Ancestor clones (*green*) containing initiating aberrations such as hyperdiploidy or primary IgH translocations occupy the available plasma cell survival niches in the bone marrow leading to monoclonal gammopathy of undetermined significance (*MGUS*). In the first phase, additional mutations result in subclones with increased fitness, and (multiple?) selective sweeps of advanced clones (*yellow*) finally replace MGUS/MM progenitors. In the second phase all available niches are occupied by advanced clones increasing the environmental constraints. At this stage the likelihood of invasion and sweeps is decreased favoring regional outgrowth of highly advanced clones (*red*). A more detailed description of this concept is provided in the main text

"Big Bang" evolutionary model[7, 36], the difficulty in replacing another dominant clone after successful invasion at that site could just mimic regionally restricted evolution and that low-frequency variants may not be detectable using our WES approach (further details in methods). However, we sequenced a subset of samples to a depth of ~ 500× at the sites of unshared mutations and confirmed the absence of >90% of these mutations at the "negative" site using a highly conservative threshold (Supplementary Data 6 and Supplementary Fig. 11). In addition, targeted sequencing also confirmed the absence of clonal BRAF (patient no. 1, 654× depth) and KRAS (patient no. 44, 371× depth) mutations in the paired "negative" samples, further supporting regionally restricted aberrations in MM. Of note, even at late stages invasion can still happen; yet, despite this we observed a trend for lower proportions of shared-diff mutations in cases with a high tumor burden; a type of heterogeneous mutation that, we interpret, indicates the presence of exchange between sites.

In Fig. 5 we illustrate these considerations in the context of a non-neutral progression model with two main phases that may explain our observations in MM patients with FLs: the first phase is characterized by selective sweeps of more advanced "fitter" clones replacing previously dominant subclones. This idea is supported by our observation that progression events such as gain of 1q are frequently ubiquitously distributed in newly diagnosed MM patients; an aberration less frequently seen at the monoclonal gammopathy of undetermined significance (MGUS) stage[17, 37]. It is also supported by recently published evidence for clonal expansion and an increase in the number of mutations during the progression from MGUS to MM[4, 17]. In the second phase advanced clones occupy all available niches increasing the environmental constraints and limiting invasion by other yet more advanced clones, resulting in the regional outgrowth of subclones. Although there may still be some exchange between sites, this phase is dominated by regional progression. The concept of spatial constraints limiting the capacity for selective sweeps was recently introduced for bacteria and solid tumors[7, 38, 39] and it is very appealing to explain our observations in MM, where BM survival niches are limited. We suggest that this type of evolutionary selective pressure could finally lead to the selection of clones with decreased BM dependence, resulting in clones crossing an evolutionary boundary, allowing for the growth of disease at extramedullary sites. Of note, while derived from patients with FLs, "regional progression" could also occur in cases without FLs as we see cases lacking FLs, that show considerable heterogeneity in space and as the number of sites we investigated was limited.

According to our interpretation, highly advanced clones growing as FLs should contain strong and more numerous driver events. Indeed, we found bi-allelic events affecting CDKN2C, RB1 and TP53, as well as other prognostically relevant chromosomal events[13, 40] in these regions, especially in large FLs (diameter >2.5 cm). Our results also indicate that the repertoire of progression events in MM has not been completely described because some FLs did not present with any of the typical MM drivers as clonal events. In this respect because they were seen and were associated with spatial heterogeneity in two patients, mutations in IL6ST and STAT3 are promising candidates as novel MM drivers. Mutations in ANK1 and MTR, which were heterogeneous in three patients each, as well as aberrations in PCLO, which were shared in six patients and recently identified as recurrently mutated in MM cell lines[41], are additional candidates. However, passenger mutations may also contribute to spatial heterogeneity due to the long evolutionary time it probably takes to become an advanced clone. This contention is highlighted by frequent site-specific non-silent mutations affecting the huge TTN gene that is not expressed in MM cells (unpublished observations). As such, even

if FLs show enrichment for unidentified driver events, it will be difficult to differentiate them from passengers.

Importantly, our approach represents a snapshot of ongoing evolution and could be considerably enhanced by using a combination of longitudinal and spatial investigations, starting at premalignant stages of the disease, to give a more complete picture of the complex evolutionary processes. A spatial-longitudinal study may also allow for a better interpretation of the results seen in treated patients. Regionally dominant clones after treatment could indicate multiple independent therapy resistant clones. Alternatively, a single minor subclone prior to therapy could be selected for. Further regional evolution of this clone could lead to the type of spatial heterogeneity seen in treated patients. The latter is supported by the observation that eight of eleven treated patients showed shared mutations in genes involved in the MAPK pathway. Whether the mutagenic effect of chemotherapy has an impact on genomic spatial heterogeneity is still elusive. However, we could not detect a signature with the classic features of alkylating agents or cisplatin exposure in our recent study of MM patients relapsing after chemotherapy[13].

We have focused on variants within tumor cells and show that these are important. However, other mechanisms could result in the development of FLs including local differences in the tumor microenvironment selecting for clones with distinct genomic aberrations[42]. Thus, in the future it will be important to investigate the interactions between tumor cells and their microenvironment in FLs and compare these to FL-free sites.

In conclusion, we show for the first time the extent of spatial heterogeneity in newly diagnosed MM. The results provide new insights in the underlying biology of MM progression and also have considerable implications for diagnostic strategies in the clinic. The non-homogeneous distribution of high-risk disease could readily result in misclassification of MM and is a likely explanation for the lack of sensitivity of currently used risk classifiers[43, 44]. Last but not least, the type of spatial heterogeneity described herein also poses a significant challenge for mutational targeted therapy[33].

## Methods

**Patients**. We performed a spatial genomic analysis of randomly collected iliac crest diagnostic samples (RNAS) and CT-guided fine needle aspirates (FNAS) from 42 newly diagnosed patients for which DNA was available (Fig. 1). The impact of treatment on spatial heterogeneity was analyzed using samples from 11 patients, of whom two were also in the set of newly diagnosed patients (Fig. 1). Patient characteristics are presented in Supplementary Table 1. The origin of samples and the type of treatment applied to the set of treated MM patients are shown in Supplementary Data 1 and 4.

The prognostic impact of site-specific high-risk clones was investigated using paired RNAS/FNAS GEP70 data from 263 patients enrolled in the TT protocols TT2- (n = 1), TT2 + (n = 2), TT3b (n = 23), TT4 (n = 156), TT5 (n = 35), or TT6 (n = 46)[3, 20, 46, 47]. Further patient characteristics are shown in Supplementary Table 1.

Informed consent for treatment and sample procurement in accordance with the Declaration of Helsinki was obtained for all cases included in this study that had been approved by the local institutional review board (no. 02815).

**Whole-exome sequencing and variant calling**. Tumor and control DNA were isolated from CD138-positive PCs and peripheral blood leukapheresis products collected after induction therapy, respectively. WES libraries were prepared using the SureSelect[QXT] sample prep kit and the SureSelect Clinical Research Exome kit (Agilent, CA, USA) with additional baits covering the Ig and MYC loci. Paired-end sequencing was performed to an average sequencing depth of 129 (range 79–230, SD 30) on a HiSeq 2500 (Illumina, CA, USA).

De-multiplexing of raw sequencing data was performed using CASAVA v.1.8.4 (Illumina). Reads were aligned to the human genome reference GRCh37 release 75 using BWA-mem version 0.7.12[48]. Sambamba v0.5.6[49] was used to sort and index bam files and to mark duplicates. Base recalibration and Indel realignment were performed according to Genome Analysis Toolkit (GATK) Best Practices v3.5[50]. Mean bait coverage, and percentage of unique reads were determined using the Picard CalculateHsMetrics tool (The Broad Institute). Somatic SNVs were called using MuTect v1.1.7[51] with default parameters. Further filtering was performed

using the fpfilter.pl script (https://github.com/ckandoth/variant-filter) with default parameters and a VAF threshold of 5%. Indels were called using Strelka v1.0.14[52] and filtered using a threshold of 5% VAF. After exclusion of variants located in immunoglobulin loci, we determined read counts for all mutations per patient using the Rsamtools R package v1.24.0 and the following inclusion criteria: unique reads, coverage exceeding 20× in all samples of the patient, a mapping quality of at least 20 and base quality of at least 20 at the site of the variant. Missense, nonsense, splice-site, and frameshift mutations were defined as non-silent.

Variants (SNVs and Indels) were classified as follows: non-ubiquitous variants and ubiquitous mutations with at least a three-fold difference in CCF between paired samples were called unshared and shared-diff, respectively. To avoid an overestimation of heterogeneity, we increased the threshold for heterogeneous mutations to a CCF of 0.2 (corresponding to a clonal proportion of 20%). For unshared mutations we also increased the threshold of total reads to ≥50 in the paired sample. Usually, MuTect only calls mutations if at least three variant reads are detected. We decreased this number to 2 (a single read was considered negative/noise). Heterogeneous mutations that did not fulfill these criteria were only added to the total number of mutations. With an average coverage of ~ 140× (range: 50–540×) at the location of unshared mutations in "negative" samples (Supplementary Fig. 1B), the sensitivity threshold for these mutations was ~1.5% (range: 0.4–4%) VAF, corresponding to a CCF of 3% (0.8–8%) for a diploid site and a tumor purity of 100%.

Ig and *MYC* translocations were called using Manta v0.20.2[53] with a variant quality score threshold of 30. Translocation calls were manually inspected in IGV[54]. SNVs, Indels and translocations we annotated using ANNOVAR[55] and SNPeff v4.2[56].

**Copy number profiling.** High-resolution HumanOmni 2.5 SNP array (Illumina) data were available for 35 newly diagnosed and 9 treated patients and pre-processed using GenomeStudio V2011.1 (http://www.illumina.com/applications/microarrays/microarray-software/genomestudio.html). CNAs were called using the ASCAT R package v2.4.3 for array data and using Sequenza[57] for WES data. For patients with array and WES data, both methods were used. The Log2 relative ratio (LogR) and the beta allele frequency (BAF) for samples without array data were calculated by applying the Battenberg algorithm (https://github.com/cancerit/cgpBattenberg) to WES data. For each sample, the accuracy of copy number calls was verified by manual inspection of LogR and BAF values for each CNA. To avoid overcalling heterogeneity and accounting for the lack of array data for a subset of patients we generally only included CNAs that were clonal in at least one of the samples and used a threshold of 1 Mb for global CNA analyses. For the detection of deletions affecting frequently mutated genes we used a threshold of 10 kb for samples with high-resolution data and 1 Mb for samples with WES data only.

**Targeted sequencing.** ≥50 ng of extracted DNA from 53 samples of our WES study (31 patients) was processed on the FoundationOne Heme (F1H) panel (Foundation Medicine, MA, USA)[58]. The current panel analyzes the complete coding DNA sequence of 405 genes. Sequencing was done to an average depth of 468× on a HiSeq 2500. Read counts at locations of mutations called by WES and covered by F1H were determined using the Rsamtools R package V1.24.0 as described for WES. The correlation between F1 and WES data is shown in Supplementary Fig. 12.

**Derivation of the GEP70 risk signature.** For risk stratification, we applied the GEP-based GEP70 model[24]. Briefly, GEP of CD138-enriched PCs was performed using Affymetrix U133Plus2.0 microarrays (Santa Clara, CA). Raw intensity values were MAS5 normalized and converted to $\log_2$ scale. Batch effect adjustment was done using M-ComBat[59]. The GEP70 corresponds to the average $\log_2$ expression of 51 upregulated genes minus the average $\log_2$ expression of 19 downregulated genes which were described previously[24]. Samples with a score ≥0.66 were assigned to the high-risk group.

**Evolution patterns and subclonal reconstruction.** The CCF was calculated as recently described[60]. Briefly, the mutation copy number was determined using the equation:

$$n_{mut} = f_s * \frac{1}{p} \left[ p n_{locus}^t + 2(1-p) \right]$$

where, $f_s$ is the observed variant allele frequency, $p$ is the tumor purity, and $n_{locus}^t$ is the locus-specific copy number as predicted by ASCAT or Sequenza. Comparing the expected $f_s$ value to values assuming the observed mutation was on 1,2,3, …, $C$ chromosomes, we assigned $n_{chr}$ the value of $C$ with the maximum likelihood using a binomial distribution. The CCF was calculated by dividing $n_{mut}$ by $n_{chr}$. Clonal substructures were inferred using SciClone[61] with default parameters and the filtered set of SNVs. For the manual design of mock phylogenetic trees, the output of SciClone was further interpreted after inclusion of copy number data, focusing on clonal mutations and CNAs.

**Application of a neutral evolution model.** To discriminate between neutral and non-neutral evolution we applied a parameter-free model for neutral growth[26] to the WES data. As recently advocated[26], samples with at least 12 SNVs with a VAF within the boundary of (0.12,0.24) were included. A linear model was fitted using the equation

$$M(f) = \frac{\mu}{\beta} \left( \frac{1}{f} - \frac{1}{f_{max}} \right)$$

where M(f) is the cumulative number of mutations per frequency, $f$ the variant allele frequency, $f$max the expected variant allele frequency of clonal mutations, and $\mu/\beta$ the mutation rate per effective cell division, which corresponds to the slope of M(f). An $R^2 \geq 0.98$ indicates neutral evolution.

**Medical imaging.** PET with CT attenuation correction (PET-CT) and diffusion-weighted magnetic resonance imaging with background suppression (DWIBS) were done within the clinical routine[45]. PET-CT was performed on a Biograph 6 PET/CT system (Siemens Medical Solutions, PA, USA), a GE Discovery IQ scanner (GE Healthcare, IL, USA) or a CTI-Reveal scanner (Siemens Medical Solutions). Images were acquired from the vertex to toes. After iterative reconstruction, images were reviewed using the PET volume computer assisted reading (VCAR) software (AW server, version 3.2, General Electric, WI, USA).

DWIBS was performed on a 1.5 Tesla Achieva scanner (PHILIPS, MA, USA). Depending on the patient's height, scanning was performed from vertex to toes in 7 to 9 slabs. A coronal whole-body T1 turbo spin echo image was obtained as a localizer. Images were analyzed in an inverted gray scale with fused whole-body maximum-intensity projection reconstructions of the diffusion and exponential apparent-diffusion coefficient images.

For PET-CT, an FL was defined as a circumscribed focus with increased FDG uptake compared to its surroundings[62]. For DWIBS, an FL was defined as a well delineated focal intensity above the surrounding background BM ≥ 1 cm in size[62]. The size of FLs corresponded to the largest diameter in cm. For 32 of 42 newly diagnosed patients DWIBS images were available and used to determine the size of FLs; for the remaining patients PET-CT images were used.

**Statistical methods.** Statistical analyses were carried out using the R software package 3.3.1. Group comparisons of continuous variables were done using the Mann-Whitney Wilcoxon test for independent groups. Subgroups with different levels of spatial heterogeneity were identified using recursive partitioning as implemented in the R package party[63]. Correlation coefficients were determined using Spearman's rank correlation. Linear regression models were fitted to investigate the relationship between the level of spatial heterogeneity and the size, the number and the anatomical distance of FLs from the iliac crest site. Correction for multiple testing was performed using the method of Bonferroni-Holm in order to control the family-wise error rate at the two-sided level of 0.05. Overall survival was defined from the initiation of protocol therapy to the date of death from any cause or censored at the date of last contact. Survival rates were estimated using the method of Kaplan-Meier. The log-rank test and the Cox proportional hazards model were used to perform group comparisons and access the impact of prognostic factors, respectively.

**Limitations.** The threshold for detection of unshared mutations and CNAs was approximately 3 and 20% CCF, respectively. Thus, a shared-diff variant with a lower frequency than these cutoffs could be "misclassified" as unshared. The CCFs of unshared mutations per patient, the read depth at the site of the "negative" paired sample and the CCF difference for shared-diff mutations are presented in Supplementary Fig. 1B–D. To account for differences in the purity of samples (Supplementary Data 1) and the coverage of WES, we used stringent cutoffs (see above). As a result, our study rather underestimates spatial heterogeneity. Moreover, the limited set of samples per patient investigated in this study, potentially also led to an underestimation of heterogeneity.

**Data availability.** The whole-exome sequencing dataset is deposited in European Genome-phenome Archive (accession: EGAS00001002111). All other data are either presented in the manuscript or available on request.

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

## Acknowledgements

We thank the patients and staff of the Myeloma Institute, UAMS. We also thank the Department of Radiology, UAMS. This work was supported by P01 CA 55819 from the National Cancer Institute. Leo Rasche was supported by the Deutsche Forschungs-gemeinschaft (DFG).

## Author contributions

Conception and design: N.W., L.R., and G.M. Provision of study material or patients: G.M., B.B., F.v.R., M.Z., S.T., C.S., and F.D. Sample management and processing: O.W.S., R.T., S.D., and P.H.P. Whole-exome sequencing: O.W.S., R.T., and P.H.P. Bioinformatic and statistical analyses: S.C., T.M., and N.W., Additional analyses: C.A., M.B., C.S., T.B., and G.M., Data interpretation: L.R., N.W., G.M., S.C., T.M., and B.W., Wrote the paper: N.W., L.R., and G.M., Reviewed and approved the paper: All authors

## Additional information

**Competing interests:** B.B. is a co-inventor on patents and patent applications related to use of GEP in cancer medicine that have been licensed to Signal Genetics Inc. The remaining authors declare no conflict of interests.

