## [Peer Review File · Nature Communications]

Reviewers' Comments:

Reviewer #1 (Remarks to the Author)

This study by Chavan et al evaluates the spatial genomic heterogeneity in myeloma by performing whole exome sequencing on biopsy specimen's from different regions. Although the sample size studied is relatively large, there are number of issues with the study. First, such spatial differences have been described in number of other malignancies and have been studied in smaller sample size in myeloma, using various technologies. In fact a recent study published in Cancer discovery and then later on in Blood shows such spatial differences and clonal evolution in different regions using targeted sequencing. Next, the interpretations, two phase model of progression and clonal sweeps, are not supported by the data. In this descriptive manuscript there is no confirmation of any of these findings. Third, clonality and subclonality of observed mutations have not been quantitated and in absence of that it will be difficult to conclude regarding clonal drift as well as sweep.

Other Major Comments:

1. It is unclear, in absence of deeper sequencing and/or PCR confirmation, whether the discordant mutations or clonal changes reported in 2 different regions represent a relative change in proportion or newer clone. Without a detailed study, and with reported variant allele frequency (VAF) of 5%, authors cannot detect a less frequent clone, and cannot conclude as such.
2. Paired end sequencing of 129X is inadequate for such analysis.
3. Fig 2A: - What is the concordance between FNAS and RNAS for cytogenetic data?
- Is there a CNA region that one method frequently missed? - What was the clonality for missed events? - Last sentence of page 4/ first sentence in page 5: Needs some number and correction based on observed events.
4. The majority of clonal differences are presented as large size cytogenetic changes. Does this mean there are no significant mutational differences between these various regions?
5. Fig 2C and 2B: - What proportion of unshared mutations were sub-clonal and how much sensitive the sequencing data to handle the sub-clonality? - Samples those have > 60% difference, do they have a common clonal potentially driver mutation - Within the same sample with different FL, how the sequencing effects the results? Is there a correlation between sequencing depth and number of mutations within the samples? - Total number of mutations was combined from multiple site or just total number of mutation in one site?
6. The differences between size and unshared mutations may reflect variation due to sampling size rather than true biological differences.
6. Line 164 - 167 says there is enrichment for relapsed patients in MAPK pathway members but fig 2d shows almost all types of mutations for those patients. Was there a statistical enrichment or this is just a claim without testing?
7. Between different sites, was there a sample purity difference? If yes, How was the analysis adjusted for that? What was the purity for each sample at multiple location?
8. Shaughnessy et. al. previously showed that patients identified as high risk by GEP70 (ranging from 15-30% of all patients, depending on the characteristics of the patient population profiled). In figure 3b combined (both high risk or high risk at one site) ~23% of patients are showed as high risk. If we take each individually what proportion of patients are considered high risk and samples those do not show concordance, how big the deviation between sites.

Reviewer #2 (Remarks to the Author)

Spatial Genomic Heterogeneity in Multiple Myeloma Revealed by Multi-Region Sequencing

Rasche and colleagues performed genomic analysis of diagnostic bone marrow biopsies and CT-guided fine needle aspirates of focal lesions in 42 newly diagnosed and 11 treated multiple myeloma (MM) patients. Chromosomal and mutational level spatial heterogeneity was found in both categories of patients in around 70% of cases. They demonstrate that some previously known initiating events such as IgH translocations are always shared; however others, such as hyperdiploidy (which is a prognostic marker) were heterogeneous, demonstrating the potential bias in single-site assessment of predictive markers. They confirm the findings reported previously by Lohr and colleagues (2014) that mutations in the genes in the MAPK pathway are frequently subclonal in MM. They demonstrate parallel evolution with respect to STAT3.

Based on the multi-region genotype, they propose a two phase pathogenic model for MM: phase 1 - clonal sweep for advanced clones in early disease stage, phase 2 - regional outgrowth of more advanced sub-clones after the all niches being occupied with "fit" clones.

Subclonal complexity has been documented in multiple myeloma in several studies to date (Lohr et al 2014; Keats et al 2012; Egan et al 2012; Bolli et al 2014) and as such is not a novel finding.

However, what had not been demonstrated to date and what is novel and exciting about this work for which the authors should be commended (assuming concerns can be addressed) is the spatial dimension of heterogeneity and regionally restricted evolution which would make this manuscript exciting to the Nature Comm readership.

Major points

1. The sequencing depth is very variable (79-230_- did the authors observe a correlation with mutational burden? Is it at the level comparable with publicly available MM datasets? Do the authors see any link between mutation burden (SNV an SCNA) and sample purity?
2. The authors choose 5% as the VAF threshold for both SNV and INDEL calls, but they do not discuss how using this hard cut off may result in misclassifying shared-diff mutations as private.
3. For copy number profiling, 35 newly diagnosed and 9 treated patients were performed SNP array and copy number was called by ASCAT. For the rest of patients, the authors used Battenberg to calculate logR and BAF, and Sequenza for CNA using WES data. How consistent is the output from the two methods? To what extent does stromal contamination affect SCNA calls?
4. Line 98 please explain how copy number were derived in the text
5. The authors used sciClone to infer clonal structure, but only Battenberg can be used to estimate subclone copy number. Does the Battenberg output agree with the phylogenetic trees based on mutations?
6. Line 162- I am not sure this statement is entirely logical- for patients that undergo a good response - a bottlenecking event would be expected to lead to less spatial heterogeneity- however for non-responders I am less convinced- also what about the mutagenic impact of therapy?
7. Although the levels of ITH between treatment naïve and treated patients were comparable this should not be taken as evidence that there is no bottlenecking. Indeed a few of the treated patients have "shared" MAPK mutations which were likely to have started off as minor subclones prior to therapy. Perhaps the authors could comment in more detail?
8. Using subclonal clustering algorithms, can the authors infer the number of subclones at each site of disease and the extent of subclone intermixing?
9. Could the authors clarify in line 147- was this evidence for parallel evolution of CDKN2C and TP53 aberrations in the same patient?
10. Line 151- are ANK1 and MTR known drivers in this disease?
11. Line 154- ditto PCLO etc
12. Line 195 and line 226- how do authors define high risk and low risk subclones?
13. Line 254- competition and cooperation could be inferred- Case 20 has evidence of two subclones occupying 4 distant sites

14. The colour-coding in Figure 2D is hard to distinguish

15. The relationship between ITH and the size of FL is intriguing but could the authors investigate this apparent link further?

Reviewer #3 (Remarks to the Author)

Rasche et al performed a spatial genomics analysis on a cohort of MM patients showing a significant heterogeneity across focal lesions, highlighting the importance of analyzing multi-region in order to better understand the intra-patient heterogeneity. The manuscript addresses a very relevant subject in MM biology that could potentially lead to an improvement in the risk stratification and personalized therapies. The manuscript is well written, the figures are clear, the experiments are done using cutting edge technology and the conclusions are supported by the data. I think it is a very relevant study that advances in the field of MM biology.

I have some few comments that need to be addressed in order to improve the clarity of the manuscript.

a) The investigators showed 2 cases where a hyperdiploid karyotype in differentially found across sites. The same applies to -13q. Both, -13q and hyperdiploid MM (H-MM) are very early and stable event in MM pathogenesis and, as long as I know, there are no clear reports showing a switch from H-MM to NH-MM or vice versa. It would be interesting that the authors elaborate a little more in the similarities and differences between those biopsies. Unfortunately, the way the data is presented precludes for performing a complete comparison. Reading lines 112-120 it is not clear what is the level of similarity of the karyotypes between sites either. For example: Is it possible that the 4th lumbar vertebra of case #1 shared trisomies in odd chromosomes with the iliac crest, but subsequently loss another chromosomes leading to the change from H-MM to NH-MM? The authors should show in suppl table 3, not only the non-shared abnormalities but also the shared abnormalities between sites for better understanding of the evolutionary tree.

b) Is it possible that some of the differences between sites could be explained by the ratio of tumor/normal PCs collected in the guided versus non-guided aspirates? A priori, guided aspirates should be richer in tumor cells. On the other hand, it could be possible that the non-guided aspirates are diluted with normal PCs, which would affect the detection of abnormalities. That would be especially affecting the copy-number changes detections, considering the relatively low sensitivity offered by copy-number arrays. How did the authors measure the tumor purity in each sample? Have the authors validated the lack of CNAs by an independent approach, such as FISH?

c) The significant reduction of differences in the pre-treated samples compared with the newly diagnosed is an interesting observation reinforcing the evolutionary dynamics and clonal evolution of MM. The authors should discuss those observations in more detail.

d) Missing a thoughtful discussion of the lower heterogeneity in relapsed disease

e) Are the 4 cases showed in figure 4 the only cases with multiple FLs analyzed or additional cases were analyzed? Please clarify.

f) Figure 4 is very nice. The idea of showing color variations for the different subclones makes sense but it makes a little more difficult the visualization of the medical images (especially fig 4A). I suggest changing the color codes of the subclones showed in fig 4A, C, and D.

g) Supplementary tables 1, 2, 4 and 6 have format issues in the pdf version. Please correct

Response to Reviewer #1

COMMENTS:

1. This study by Chavan et al evaluates the spatial genomic heterogeneity in myeloma by performing whole exome sequencing on biopsy specimen's from different regions. Although the sample size studied is relatively large, there are number of issues with the study. First, such spatial differences have been described in number of other malignancies and have been studied in smaller sample size in myeloma, using various technologies. In fact a recent study published in Cancer discovery and then later on in Blood shows such spatial differences and clonal evolution in different regions using targeted sequencing.

Response:

A) This study by Rasche *et al.* is the first systematic genomic study of spatial heterogeneity in this hematological cancer revealing an unexpected high level of regional clonal dominance. As written by reviewer #2 and #3 it “addresses a very relevant subject in MM biology that could potentially lead to an improvement in the risk stratification and personalized therapies” and shows “the spatial dimension of heterogeneity and regionally restricted evolution”.

B) We have extensively reviewed the literature and to the best of our knowledge not a single systematic approach has been published for myeloma. The two studies that used modern molecular methods (targeted sequencing and FISH, respectively) and investigated paired samples collected at the same time were presented as case reports (Raab *et al.* Blood 2016 & López-Anglada *et al.* EJM 2010^{1,2}). The study by Raab *et al.*, which was already cited in our discussion, was limited to the description of two genes, *BRAF* and *NRAS*, and showed three different *NRAS* mutations in three different focal lesions in a patient with late stage relapsed-refractory disease. Since *NRAS* mutations may overcome the effect of *BRAF* inhibition and the mutations were not detectable at inclusion of the patient into *BRAF* treatment, this heterogeneity in space was probably “triggered by treatment”. The authors concluded “Spatially divergent clonal evolution triggered by therapy occurs in this hematologic malignancy (...)”. This study also did not show spatial heterogeneity for *BRAF* mutations, in contrast to our study. The fluorescence in situ hybridization study by López-Anglada *et al.* presented one newly diagnosed MM patient with a del(17p) only detectable in extramedullary disease. We have cited this paper in the revised version of our manuscript. As stated by the authors EMD rather represents end stage disease. As such, this study may not be representative for newly diagnosed patients. We have modified the discussion:

“This multi-region genomic analysis is the first systematic study that describes spatial heterogeneity in MM; a phenomenon that has been recently shown for several solid cancers³⁻⁹. Furthermore, our observations are supported by two case reports describing spatial heterogeneity in one MM patient with extra-medullary disease² and in one relapse-refractory MM patient treated with the *BRAF* inhibitor Vemurafenib¹.”

A third study published in 2001 by the UAMS myeloma team¹⁰ analyzed random aspirates and paired focal lesions using conventional cytogenetics. Results from metaphases cannot be considered appropriate to analyze genomic alterations in MM due to the low proliferation rate of MM cells. Furthermore, the authors solely classified the samples as positive or negative for metaphases. Two other recent studies^{11,12} investigated circulating MM cells and circulating tumor DNA along with paired bone marrow samples, respectively. In both investigations the authors found mutations in the peripheral blood, which were not detectable in the paired bone marrow or were present at significantly lower frequencies. Based on their results the authors concluded that spatial heterogeneity was present in MM. However, since the (exact) origin and nature of circulating tumor cells and DNA remains elusive, the results only indicated spatial heterogeneity. Thus, our study is the first to present spatial heterogeneity within the skeletal system in newly diagnosed MM patients.

C) We used high resolution copy number arrays and whole exome sequencing to investigate the spatial clonal architecture, as well as gene expression based risk scores and patients' clinical outcome data to show the potential impact of this heterogeneity on risk stratification. By combining genomic data with imaging information we show for the first time that extensive genomic heterogeneity in space is even reflected in imaging patterns.

In summary, this type of comprehensive approach has not been performed before.

2. Next, the interpretations, two phase model of progression and clonal sweeps, are not supported by the data. In this descriptive manuscript there is no confirmation of any of these findings.

Response:

A) We interpreted our results and put them into the context of previous research. The widely accepted current model of MM evolution includes the concept of initiating events, ongoing mutational processes and competition between clones with different fitness levels as described in several recent state-of-the-art review articles on MM evolution (Morgan 2014 Nat Rev Cancer¹³, Ghobrial, 2016 Nat Rev Clin Oncol¹⁴). This Darwinian type model "is based on the idea that mutations are acquired randomly and are selected based on the clonal advantage they confer"¹³. The dynamics of Darwinian models of tumor evolution are nicely described by Robertson-Tessi and Anderson in their recent *Nature Genetics* "New and views" article: during "clonal selection sequential mutations lead to fitter clones that sweep through the population"¹⁵. Since our data do not support a "neutral evolution" model, we have extended the "selection" model to account for our observations.

B) The first phase corresponds to the accepted **simple** Darwinian model. Clonal "sweeps" are actually also supported by our data, since we frequently observed shared "progression" events in clones with unshared mutations (complex trunk). This was already stated in the discussion of the old version of our manuscript: "This idea is supported by the frequent observation of ubiquitously distributed progression events such as gain of 1q, aberrations not frequently seen at the monoclonal gammopathy of undetermined significance (MGUS) stage". This is also supported by our observation of "non-neutral" evolution in our set of newly diagnosed patients as determined using an accepted equation to discriminate between a neutral and a Darwinian type of evolution³, and the high frequency of shared "progression" events in our set of treated patients. The idea of a second phase that is characterized by "regional evolution" is based on our observation of unshared "progression" events, regional clonal dominance, and a trend to a lower frequency of shared-diff mutations in ISS3 patients and patients with a deletion at 1p.

C) Regional evolution in end stage and heavily treated patients is also supported by the case reports of Raab *et al.* Blood 2016 & López-Anglada *et al.* EJM 2010.

D) However, it is still a proposed model and we appreciate that. As already written in our discussion “our approach represents a snapshot of ongoing evolution and could be considerably enhanced by using a combination of longitudinal and spatial investigations to give a more complete picture of the underlying evolutionary processes.”

We modified the discussion and wrote:

“As one possible interpretation of previous results and our own observations, we propose an extended pathogenic model for MM with two main phases (...). AND “In our proposed model, sub-clones growing in a nodular fashion in the BM represent an intermediate stage”

E) As written by reviewer #3 “the manuscript is well written, the figures are clear, the experiments are done using cutting edge technology and the conclusions are supported by the data.”

3. Third, clonality and subclonality of observed mutations have not been quantitated and in absence of that it will be difficult to conclude regarding clonal drift as well as sweep.

Response:

We thank the reviewer for the suggestion to improve the description of sub-clonal vs. clonal events. In the new version of Figure 2C and 2D sub-clonal and clonal unshared mutations are presented in different colors.

4. It is unclear, in absence of deeper sequencing and/or PCR confirmation, whether the discordant mutations or clonal changes reported in 2 different regions represent a relative change in proportion or newer clone. Without a detailed study, and with reported variant allele frequency (VAF) of 5%, authors cannot detect a less frequent clone, and cannot conclude as such.

Response

A) One of our interpretations/conclusions was that *advanced* clones grew in (large size) focal lesions. This interpretation was not solely based on the frequency of heterogeneous mutations but also on the frequent observation of complex genomic aberrations and unshared driver events in regionally dominant clones, the frequent observation of clonal and sub-clonal unshared events at the same site, and the association between the size of focal lesions and the proportion of unshared events. Both unshared and shared-diff mutations would be well explained by regional evolution. Of note, we do not exclude exchange between sites but rather a limitation at later stages of the disease.

B) In our opinion, a higher coverage will not necessarily allow for a discrimination between a relative change in proportion or newer clone, because even the absence of a clone at one site could still be due to a change in proportion at that site.

C) Regarding the threshold for detection of unshared mutations, the corresponding methods paragraph was obviously difficult to understand. We apologize for that. As a first step, we used a 5% VAF cut-off for detection of mutations and Indels. To avoid an overestimation of heterogeneity we increased the threshold for heterogeneous mutations to a cancer clonal fraction of 0.2 (corresponding to a clonal proportion of 20%). For unshared mutations we also increased the threshold of total reads to ≥ 50 in the paired sample. Usually, MuTect only calls mutations if at least 3 variant reads are detected. We decreased this number to 2 (a single read was considered

negative/noise). The median number of total reads for unshared mutations in the paired “negative” sample was 142. Thus, the reported variant allele frequency for *unshared* mutations in the “negative” paired sample was on average less than 1.5%.

D) Usually, a CT-guided FL sample typically consists of a 2 ml cell suspension containing a limited number of cells. Since gene expression profiling analysis have been prioritized for decades in our institute and samples have not routinely been processed using RNA/DNA columns, sample availability for our study was limited and was further restricted by the use of two independent methods for CNA calling. However, for ~three years our institute has been sending selected samples to FoundationOne (Cambridge, MA) for targeted sequencing of 405 genes. We have data for 31 patients and 53 samples that were included in our study. Since the FoundationOne data was incomplete we did not report on target sequencing in the first draft of the paper. However, to prove that our results are adequate for a statistical analysis of quantitative data, we have updated our manuscript and included FoundationOne data. Usually, FoundationOne reports on variants with a variant allele frequency ≥ 0.05 only. Mutations are reported down to 0.01 where the variant is a known hotspot and there is sufficient purity and sequencing depth. To generally allow for a higher sensitivity of mutation calling, we have analyzed the corresponding bam files and called mutations that were found by our WES approach and were located in genes covered by the FoundationOne set. Of note, the R^2 was 0.94 for a comparison of variant allele frequencies of 264 mutations (please see the new Supplemental Figure 7). Of 12 mutations that were negative according to WES data, 10 were also negative according to FoundationOne data. The two “positive” variants showed a variant allele frequency of 0.008 and 0.01. Thus, we assume that the majority (>80%) of *unshared* mutations indeed had a variant allele frequency of less than 1.5%. This threshold is similar to the threshold used for a landmark paper by Gerlinger *et al.* who investigated spatial heterogeneity in clear cell renal cell carcinoma and called variants if the allele frequency was 0.5-1% in targeted sequencing data¹⁶. Furthermore, three of the negative variants in our set were *BRAF* (654 reference reads, clonal in paired sample, patient #1), *KRAS* (371 reference reads, clonal in paired sample, patient #42), and *ATR* (508 reference reads, CCF of 0.4 in paired sample, patient #11) mutations. FoundationOne data also confirmed the unshared bi-allelic deletion of *TP53* in patients #1. Together, we are confident that the vast majority of mutations described in our manuscript are valid and that *unshared* mutations indeed showed at least a huge frequency differences between paired samples or were even “real” unshared events. However, in order to be more precise in the manuscript, we have updated the definition of unshared variants in our manuscript and explicitly mentioned the 1.5% threshold.

We modified the methods section:

“To avoid an overestimation of heterogeneity, we increased the threshold for heterogeneous mutations to a cancer clonal fraction of 0.2 (corresponding to a clonal proportion of 20%). For unshared mutations we also increased the threshold of total reads to ≥ 50 in the paired sample. Usually, MuTect only calls mutations if at least 3 variant reads are detected. We decreased this number to 2 (a single read was considered negative/noise). Heterogeneous mutations that did not fulfill these criteria were only added to the total number of mutations. With an average coverage of $\sim 140x$ at the location of *unshared* mutations in “negative” samples, the sensitivity threshold for these mutations was $\sim 1.5\%$ VAF.”

Supplemental Figure 7: Comparison of whole exome and targeted sequencing data. 53 samples of our whole exome sequencing study (31 patients) were also processed using the targeted sequencing F1H Panel (Foundation Medicine, MA) which covers 405 genes. The plot shows the F1H and whole exome sequencing variant allele frequency for mutations that were called by whole exome sequencing and were covered by the F1H Panel.

We included a paragraph on targeted sequencing to the methods:

Targeted Sequencing

≥ 50 ng of extracted DNA from 53 samples of our WES study (31 patients) was processed on the FoundationOne Heme (F1H) panel (Foundation Medicine, MA) [REF]. The current panel analyzes the complete coding DNA sequence of 405 genes. Sequencing was done to an average depth of 468x on a HiSeq 2500. Read counts at locations of mutations called by WES and covered by F1H were determined using the Rsamtools R package V1.24.0 as described for WES.

5. Paired end sequencing of 129X is inadequate for such analysis.

Response:

A) The aim of this study was to systematically investigate spatial genomic heterogeneity in Multiple Myeloma using rather unbiased methods, e.g. whole exome instead of targeted sequencing with a limited set of genes.

B) Our coverage is within the range of reported coverages in other landmark publications of MM genomics, e.g. Lohr *et al.*, *Cancer Cell* 2014 (89x)¹⁷; Bolli *et al.*, *Nat comm* (236x)¹⁸, and Walker *et al.*, *JCO* 2015 (60x)¹⁹, Weinhold *et al.*, *Blood* 2016 (118x)²⁰. In the two longitudinal studies by Bolli *et al.* and Weinhold *et al.*, whole exome data was used to describe patterns of evolution. The branching pattern is analogous to a spatial pattern of *unshared* variants.

C) The average coverage of our study (142x at the location of unshared mutations) and our definition of *unshared* mutations resulted in an average detection threshold of 1.5% variant allele frequency which we consider adequate. As outlined above in the landmark paper on spatial genomic heterogeneity published in the *Nature Genetics* 2014 by Gerlinger *et al.* the authors used a cutoff of ~1% for *unshared* variants¹⁶.

D) The strong association of the proportion of unshared mutations with the size of investigated FLs also shows that this threshold is biologically meaningful (this association is further discussed in response to point #9), further supporting that our coverage is adequate for such an analysis. We called mutations *unshared* because it best describes the nature of these events. We agree that our coverage does not allow to fully exclude the possibility that in our “negative” samples a few cells were positive for the respective “unshared” mutation and in the manuscript we even do not exclude a certain degree of exchange between sites. To be more precise we have modified the discussion:

“We appreciate that, in analogy to the *neutral* “Big Bang” evolutionary model, the difficulty in replacing another dominant clone after successful invasion at that site could just mimic regionally restricted evolution and that low frequency variants may not be detectable using our WES approach (further details in methods).”

E) High levels of circulating tumor cells are frequently detected in myeloma (up to 1% of peripheral blood mononuclear cells) and may “contaminate” bone marrow samples. Since ultra-deep sequencing of bone marrow samples may be confounded by these cells, the value of a significantly higher coverage is questionable.

6. Fig 2A: - What is the concordance between FNAS and RNAS for cytogenetic data? Is there a CNA region that one method frequently missed? - What was the clonality for missed events? - Last sentence of page 4/ first sentence in page 5: Needs some number and correction based on observed events.

Response:

A) We have updated Supplemental table 3 and included shared and unshared CNAs.

B) To account for the difference in the resolution for detection of CNAs and coverage between DNA arrays and whole exome sequencing, we used a threshold of 1 Mb for the global CNA analysis and only included CNAs that were clonal in at least one of the paired samples. We also excluded the sex chromosomes. Using this strategy, no CNAs detectable by one method and confirmed by manual inspection were missed by the other one. Basically, both methods cannot detect CNAs in centromeric regions. However, if both the p and the q arm showed the same copy number, we assumed the same copy number for the centromeric region.

C) All presented CNAs were clonal in at least one sample. We also detected CNAs that were sub-clonal only, but due to the lower sensitivity of DNA arrays we did not include them into the comparison. We have updated the method paragraph to explain this strategy in more detail.

D) We have updated the last sentence on page 4 and added numbers:

“Changes on chromosome 1 and 4 were the most frequent contributors to spatial heterogeneity (n=7 patients), followed by chromosome 5 and 8 (n=6) including deletions, gains and LOH.”

7. The majority of clonal differences are presented as large size cytogenetic changes. Does this mean there are no significant mutational differences between these various regions?

Response:

We assume that the reviewer refers to Figure 4. By checking our manuscript we realized that an important information was missing. We focused on clonal events in the analysis of multi-regional evolutionary events. Thus, all variants (except the sub-clonal *KRAS* mutation in patient #20, Figure 4d) were clonal shared or clonal unshared events. Furthermore, gene symbols presented

in the nodes correspond to potential drivers in myeloma. Digits indicate the total number of branch-specific clonal mutations. We have updated our manuscript accordingly and added to the methods section:

“For the manual design of mock phylogenetic trees, the output of SciClone was further interpreted after inclusion of copy number data, focusing on clonal mutations and CNAs.”

8. Fig 2C and 2B: - What proportion of unshared mutations were sub-clonal and how much sensitive the sequencing data to handle the sub-clonality? –Samples those have > 60% difference, do they have a common clonal potentially driver mutation
 - Within the same sample with different FL, how the sequencing effects the results?
 - Total number of mutations was combined from multiple site or just total number of mutation in one site?
 - Is there a correlation between sequencing depth and number of mutations within the samples?

Response:

A) As requested by the reviewer, we have updated Figure 2c and 2d and used a different color code for sub-clonal mutations. Please also see our response to point #3. The median coverage at the site of unshared mutations in “negative” samples, that were sub-clonal in the positive sample, was 143x.

B) We thank the reviewer for the question whether patients with more than 60% differences have specific aberrations in common. Interestingly, they did not share a specific mutation or CNA. However, they had in common that they showed heterogeneity for at least one of the recently identified strong drivers of myeloma progression. Three of these 5 patients showed an inactivation of the tumor suppressor *TP53* or *RB*, which was only detectable in FLs. Patient #2 and #3 presented with shared-diff mutations for the MAPK genes *NRAS* or *KRAS*. To illustrate this lack of a common “driver” and heterogeneity in progression events, we have updated Figure 5.

C) The total number of mutations corresponds to all mutations found in a patient with a VAF \geq 5% (all sites).

D Calculating a linear model, we did not see a significant association between the sequencing depth and the mutation burden. This also holds true for patients with multiple focal lesion samples. This may be explained by our strategy that already accounted for differences in purity and coverage: we used a 5% variant allele frequency threshold for total mutations and 20% cancer clonal fraction for heterogeneous mutations.

9. The differences between size and unshared mutations may reflect variation due to sampling size rather than true biological differences.

Response

As described in results the rho value for a Spearman’s correlation between the size of investigated focal lesions and the proportion of *unshared* mutations was 0.62 ($P=0.000009$). Whereas 10 of 15 patients with large FLs showed more than 20% unshared mutations, none of the patients with

small (<1 cm) or without FLs presented with such a high value. According to a linear model, size explains 35% of the variance in the proportion of unshared mutations, a remarkably high value. We agree that a higher sample size may lead to a better estimate of the effect size, however we do not think that our observation was just due to random variation.

10. Line 164 - 167 says there is enrichment for relapsed patients in MAPK pathway members but fig 2d shows almost all types of mutations for those patients. Was there a statistical enrichment or this is just a claim without testing?

Response:

We appreciate that the term “enrichment” in this context was misleading. We aimed to state, that all treated patients presented with at least one of the “progression” markers/driver events in myeloma. We therefore changed the sentence to:

“All patients presented with adverse prognostic markers on chromosome 1 and all but two had a mutation in genes of the MAPK pathway, indicating a strong selective pressure of treatment (Fig. 2d).”

11. Between different sites, was there a sample purity difference? If yes, How was the analysis adjusted for that? What was the purity for each sample at multiple location?

Response:

Using values as predicted by Sequenza, there was a slight purity difference between iliac crest (median 89%) and focal lesion (median 95%) samples. We have added the corresponding values to Supplemental Table 2. To account for that we used these values to calculate the cancer clonal fraction of mutations. In addition, we only included CNAs that were clonal at least at one site and increased the CCF threshold for heterogeneous mutations to 0.2. We neither detected a significant association between the purity and the number of detectable mutations and CNAs nor the purity and the proportion of unshared mutations (linear model). Furthermore, the association between the size of FLs and unshared variants was not impacted by purity or coverage (linear model). We included a paragraph in the manuscript that describes the limitations of our study. However, these limitations do not impact our general conclusion on spatial heterogeneity in myeloma.

Limitations

The threshold for detection of *unshared* mutations and CNAs was approximately 1.5% and 20%, respectively. Thus, a *shared-diff* variant with a lower frequency than these cut-offs would be “misclassified” as *unshared*. To account for differences in the purity of samples (Supplemental Table 2) and the coverage of WES, we used stringent cut-offs (see above). As a result, our study rather underestimates spatial heterogeneity. Moreover, the limited set of samples per patient investigated in this study, potentially also led to an underestimation of heterogeneity.

12. Shaughnessy et. al. previously showed that patients identified as high risk by GEP70 (ranging from 15-30% of all patients, depending on the characteristics of the patient population profiled). In figure 3b combined (both high risk or high risk at one site) ~23% of patients are showed as high risk. If we take each individually what proportion of patients are considered high risk and samples those do not show concordance, how big the deviation between sites.

Response:

According to a recent review article by Shaughnessy *et al.* "several large clinical trials have shown that a (GEP70) high-risk signature is present in approximately 15% of new cases of MM"²¹. For newly diagnosed patients treated in the German GMMG studies we recently determined a proportion of ~11% GEP70 high risk²². In this study of 263 newly diagnosed patients with paired samples, we identified 28 cases with high risk according to both samples, 14 with high risk only according to the iliac crest sample and 20 with high risk only according to the fine needle aspirate sample. We have added these numbers to the revised manuscript. Based on these numbers, our set of patients is representative for the distribution of GEP70 scores for iliac crest samples from newly diagnosed patients. 42 patients (16%) would have been classified as high risk according to a randomly collected iliac crest sample and 20 (8%) would have been misclassified. The average deviation between sites was 0.52.

Response to Reviewer #2

Rasche and colleagues performed genomic analysis of diagnostic bone marrow biopsies and CT-guided fine needle aspirates of focal lesions in 42 newly diagnosed and 11 treated multiple myeloma (MM) patients. Chromosomal and mutational level spatial heterogeneity was found in both categories of patients in around 70% of cases. They demonstrate that some previously known initiating events such as IgH translocations are always shared; however others, such as hyperdiploidy (which is a prognostic marker) were heterogeneous, demonstrating the potential bias in single-site assessment of predictive markers. They confirm the findings reported previously by Lohr and colleagues (2014) that mutations in the genes in the MAPK pathway are frequently subclonal in MM. They demonstrate parallel evolution with respect to STAT3.

Based on the multi-region genotype, they propose a two phase pathogenic model for MM: phase 1 - clonal sweep for advanced clones in early disease stage, phase 2 - regional outgrowth of more advanced sub-clones after the all niches being occupied with "fit" clones.

Subclonal complexity has been documented in multiple myeloma in several studies to date (Lohr et al 2014; Keats et al 2012; Egan et al 2012; Bolli et al 2014) and as such is not a novel finding.

However, what had not been demonstrated to date and what is novel and exciting about this work for which the authors should be commended (assuming concerns can be addressed) is the spatial dimension of heterogeneity and regionally restricted evolution which would make this manuscript exciting to the Nature Comm readership.

Major points

1. The sequencing depth is very variable (79-230_- did the authors observe a correlation with mutational burden? Is it at the level comparable with publicly available MM datasets? Do the authors see any link between mutation burden (SNV an SCNA) and sample purity?

Response:

First of all, we appreciate the positive feedback to our study.

A) We agree with the reviewer that the range of sequencing depth was quite large. However, the standard deviation of the sequencing depth was only 30. We have added the SD to the methods paragraph of the revised manuscript. Calculating a robust linear model to account for the high

variance in the mutational burden in myeloma due to differences between subgroups and the treatment status (patients with a MAF translocation and treated patients have significantly higher numbers of mutations), we did not see a significant association between the sequencing depth and the mutation burden. The same holds true for sample purity as determined using Sequenza and mutational burden or the number of copy number aberrations. This may be explained by our strategy that already accounted for differences in purity and coverage: we included only CNAs that were clonal in at least one of the paired samples, and we used a 5% variant allele frequency threshold for total mutations, and 20% cancer clonal fraction for heterogeneous mutations.

B) Our coverage is within the range of reported coverages in other publications of MM genomics, e.g. Lohr *et al.*, Cancer Cell 2014 (89x)¹⁷; Bolli *et al.*, Nat comm 2014 (236x)¹⁸, Walker *et al.*, JCO 2015 (60x)¹⁹, Weinhold *et al.*, Blood 2016 (118x)²⁰.

2. The authors choose 5% as the VAF threshold for both SNV and INDEL calls, but they do not discuss how using this hard cut off may result in misclassifying shared-diff mutations as private.

Response:

A) Obviously, our strategy to define unshared mutations was difficult to understand. Initially, we used a 5% VAF cut-off for detection of variants. To avoid an overestimation of heterogeneity we increased the threshold for heterogeneous mutations to a cancer clonal fraction of 0.2 (corresponding to a clonal proportion of 20%). For unshared mutations we also increased the threshold of total reads to ≥ 50 in the paired sample. Usually, MuTect only calls mutations if at least 3 variant reads are detected. We decreased this number to 2 (a single read was considered negative/noise). Thus, our threshold was $\sim 1.5\%$ VAF. We have updated the description of this strategy in the methods paragraph.

“Variants (SNVs and Indels) were classified as follows: non-ubiquitous variants and ubiquitous mutations with at least a three-fold difference in cancer clonal fraction (CCF) between paired samples were called *unshared* and *shared-diff*, respectively. To avoid an overestimation of heterogeneity, we increased the threshold for heterogeneous mutations to a cancer clonal fraction of 0.2 (corresponding to a clonal proportion of 20%). For unshared mutations we also increased the threshold of total reads to ≥ 50 in the paired sample. Usually, MuTect only calls mutations if at least 3 variant reads are detected. We decreased this number to 2 (a single read was considered negative/noise). Heterogeneous mutations that did not fulfill these criteria were only added to the total number of mutations. With an average coverage of $\sim 140x$ at the location of *unshared* mutations in “negative” samples, the sensitivity threshold for these mutations was $\sim 1.5\%$ VAF.”

B) However, we appreciate that even at this lower threshold shared-diff mutations may be classified as unshared mutations. Thus, we have updated the respective sentence in the discussion:

We appreciate that, in analogy to the *neutral* “Big Bang” evolutionary model^{17,34}, the difficulty in replacing another dominant clone after successful invasion at that site could just mimic regionally restricted evolution and that variants with a $VAF < 1.5\%$ may be misclassified using our WES approach.

We also included a paragraph “Limitations” to the methods:

“Limitations

The threshold for detection of *unshared* mutations and CNAs was approximately 1.5% and 20%, respectively. Thus, a *shared-diff* variant with a lower frequency than these cut-offs would be “misclassified” as *unshared*. To account for differences in the purity of samples (Supplemental Table 2) and the coverage of WES, we used stringent cut-offs (see above). As a result, our study rather underestimates spatial heterogeneity. Moreover, the limited set of samples per patient investigated in this study, potentially also led to an underestimation of heterogeneity.”

3. For copy number profiling, 35 newly diagnosed and 9 treated patients were performed SNP array and copy number was called by ASCAT. For the rest of patients, the authors used Battenberg to calculate logR and BAF, and Sequenza for CNA using WES data. How consistent is the output from the two methods? To what extent does stromal contamination affect SCNA calls?

Response:

A) To account for the sensitivity of SNP arrays and WES data for detection of sub-clones (~20%) we only included CNAs that were clonal in at least one the paired samples. To further account for the fact that we did not have array data for all patients, we run ASCAT and Sequenza for patients with array data and by comparing them determined 1 Mb as a reliable threshold for the global analysis of CNAs. To avoid an overestimation of heterogeneity and account for the detection threshold “issue” and purity differences, we manually checked every chromosomal profile, especially at locations of CNAs that were only detected in one of the paired samples by ASCAT and/or Sequenza. “Undetectable” CNAs that actually presented as minor sub-clones were classified as shared and presented as a minor sub-clone in Supplemental Table 3. Since the visualization of the BAF by Sequenza is often difficult to interpret, especially for sub-clonal events, we also calculated logR and BAF values using Battenberg which shows these values comparable to the ASCAT output.

B) There was no significant association between the number of total or unshared CNAs and purity. Since we only included clonal/major CNAs, we expected this “negative” result for the total number. However, we cannot exclude that in samples with a significant stromal contamination, the detection of minor sub-clones was limited.

4. Line 98 please explain how copy number were derived in the text

Response:

We updated the manuscript and wrote in the corresponding results paragraph:

“We used high resolution SNP arrays and WES data to call copy number aberrations (CNAs). To account for the sensitivity of this approach for detection of sub-clones (threshold: ~20%) we only included CNAs that were clonal in at least one of the paired samples.”

5. the authors used sciClone to infer clonal structure, but only Battenberg can be used to estimate subclone copy number. Does the Battenberg output agree with the phylogenetic trees based on mutations?

Response:

A) We used Battenberg to facilitate the detection of sub-clones and thereby the identification of “false-positive” *unshared* CNAs. We initially used PhyloWGS for inference of trees for patients with multiple samples. However, the output was not satisfactory with numerous incomprehensible or even contrasting solutions. Thus, we decided to manually design these trees. First, we calculated cancer clonal fractions of mutations based on the variant allele frequency, the purity predicted by Sequenza and the main copy number at the respective site. Next, we ran SciClone using cancer clonal fractions of mutations to visualize the clonal substructure in paired samples. Since the idea was to illustrate the “main” events and branches resulting in spatial heterogeneity in patients with large focal lesions, we designed the phylogenetic trees using clonal events only. Therefore, all variants presented in Figure 4 were clonal shared or clonal unshared events except the sub-clonal *KRAS* mutation in patient #20 (Figure 4d). We included the *KRAS* clone to illustrate that an advanced sub-clone can be ubiquitously present, further supporting the complexity of clonal evolution in this disease. Furthermore, we excluded unshared mutations that were located in regions affected by an unshared deletion or loss of heterozygosity (LOH) in the paired sample, because our data did not allow to determine the order of such events. Due to this strategy, the output of Battenberg based on clonal and sub-clonal events would differ but the main branches would (should) be the same. We appreciate, that our strategy to build these trees was not sufficiently explained in the first version of our manuscript and we have modified it accordingly.

B) For the description of regional clonal dominance in the paragraph “Non-neutral evolution in Multiple Myeloma”, we used the SciClone output only. Since sub-clonal chromosomal events or the “loss” of a clonal mutation due to a deletion or LOH would not affect our observation that frequently different clones dominated at different sites we did not consider sub-clonal CNAs or correct the output for deletions at mutational sites. However, we agree, that for a detailed sub-clonal analysis, estimates of sub-clone copy numbers would be required.

6. Line 162- I am not sure this statement is entirely logical- for patients that undergo a good response – a bottlenecking event would be expected to lead to less spatial heterogeneity- however for non-responders I am less convinced- also what about the mutagenic impact of therapy?

Response:

A) We thank the reviewer for this comment. We agree that non-responders/ patients with primary refractory disease would probably show no changes in their clonal substructure. However, all patients in our analysis showed an initial response. We added the following sentence to the manuscript to be more precise:

“Thus, for initially responding patients we would expect relapse to be dominated by a limited number of highly resistant, selected clones and as a result less spatial heterogeneity.”

B) We also agree with the reviewer that chemotherapy could potentially cause harmful DNA damage and by increasing the likelihood of additional driver abnormalities lead to detectable spatial heterogeneity. However, we could not detect a signature with the classic features of alkylating agents or cisplatin exposure in our recent study of MM patients relapsing after dose-intense chemotherapy²⁰. We have updated the discussion accordingly:

“Whether the mutagenic effect of chemotherapy has an impact on genomic spatial heterogeneity is still elusive. Potentially, it could increase the likelihood of additional driver mutations by causing DNA damage. Thus, it could lead to the appearance and outgrowth of fitter clones and thereby increased spatial heterogeneity. However, we could not detect a signature with the classic

features of alkylating agents or cisplatin exposure in our recent study of MM patients relapsing after chemotherapy²⁰.”

7. Although the levels of ITH between treatment naïve and treated patients were comparable this should not be taken as evidence that there is no bottlenecking. Indeed a few of the treated patients have “shared” MAPK mutations which were likely to have started off as minor subclones prior to therapy. Perhaps the authors could comment in more detail?

Response:

A) We agree, and modified the results section:

“The average level of heterogeneity on the chromosomal and the mutational level did not significantly deviate from the corresponding values in newly diagnosed patients (Wilcoxon tests, $P>0.05$). Together, these results indicate a strong selective pressure of treatment and further regional evolution of selected clones.”

B) We have also updated the discussion:

“This may also allow for a better interpretation of the results seen in treated patients. Regionally dominant clones after treatment could indicate multiple independent resistant clones. Alternatively, a minor sub-clone prior to therapy could be selected for. Further regional evolution of this clone could lead to the type of spatial heterogeneity seen in treated patients. The latter is supported by the observation that 8 of 11 treated patients showed shared mutations in MAPK genes.”

8. Using subclonal clustering algorithms, can the authors infer the number of subclones at each site of disease and the extent of subclone intermixing?

Response

We agree with the reviewer that the number of sub-clones and the extent of intermixing at each each would be interesting. We used SciClone to visualize the clonal substructure in paired samples and thereby to facilitate the analysis of heterogeneity. In our opinion, for paired samples the SciClone output can be used to determine major clones at different sites and assign sub-clones to them. For our study, we have used these results to determine the number of patients with different dominant clones at different sites (n=25). Unfortunately, according to our experience, including single cell approaches, SciClone and other clustering algorithms cannot be used to reliably quantitate the number of minor sub-clones (VAF<20%), especially for the ones that are only detectable at one site or show similar frequencies in paired samples. Thus, we cannot use a clustering algorithm to properly infer the number of (minor) sub-clones in paired samples of myeloma. However, the number of clonal shared and shared-diff mutation per sample for newly diagnosed patients is shown in Supplemental Table XX.

9. Could the authors clarify in line 147- was this evidence for parallel evolution of CDKN2C and TP53 aberrations in the same patient?

Response: The events were detectable in 3 different patients. We modified the respective sentence to:

“Importantly, we found 3 patients with unshared ultra-high-risk bi-allelic events affecting the tumor-suppressor genes *CDKN2C* (n=1) and *TP53* (n=2).”

10. Line 151- are ANK1 and MTR known drivers in this disease?
11. Line 154- ditto PCLO etc

Response to 10 & 11:

To the best of our knowledge non-silent mutations in ANK1 and MTR have not been described as drivers in MM yet. Mutations in *PCLO* have previously been described as recurrent events in MM²³. However, the functional role of them in myeloma is still elusive. We have added a paragraph on potential novel drivers in myeloma to the discussion:

“According to this model highly advanced clones growing as FLs should contain strong and more numerous driver events. Indeed, we found bi-allelic events affecting *CDKN2C*, *RB1* and *TP53*, as well as other prognostically relevant chromosomal events in these regions, especially in large FLs (diameter > 2.5 cm). Our results also indicate that the repertoire of progression events in MM has not been completely described because some FLs did not present with any of the typical MM drivers as clonal events. In this respect because they were seen and were associated with spatial heterogeneity in two patients, mutations in *IL-6ST* and *STAT3* are promising candidates as novel MM drivers. Mutations in *ANK1* and *MTR*, which were heterogeneous in 3 patients each, as well as aberrations in *PCLO*, which were shared in 6 patients and recently identified as recurrently mutated in MM, are additional candidates. However, passenger mutations may also contribute to spatial heterogeneity due to the long evolutionary time it probably takes to become an advanced clone. This contention is highlighted by frequent site-specific non-silent mutations affecting the huge *TTN* gene that is not expressed in MM cells (unpublished observations). As such, even if FLs show enrichment for unidentified driver events, it will be difficult to differentiate them from passengers. Of note, we have focused on variants within tumor cells and show that these are important, however we do not exclude a contribution from the micro-environment.”

12. Line 195 and line 226- how do authors define high risk and low risk subclones?

Response:

We thank the reviewer for this comment. We agree that the description in the manuscript was confusing since we described high risk chromosomal markers and the risk status according to the gene expression profiling based GEP70 score. We have modified the manuscript to better distinguish between classifications according to chromosomal markers and the GEP70 score.

13. Line 254- competition and cooperation could be inferred- Case 20 has evidence of two subclones occupying 4 distant sites

Response:

We agree with the reviewer that our observations may be due to competition and/or cooperation. Since our data cannot be used to distinguish between these two types of *non-neutral* evolution we modified the sentence to:

“The evolution patterns described above strongly indicate *non-neutral* evolution.”

14. The colour-coding in Figure 2D is hard to distinguish

Response:

We agree, particularly light grey and light blue were hard to distinguish. Therefore, we merged mutation/deletion and bi-allelic deletions to “bi-allelic events” and additionally used crossed boxes

for shared “bi-allelic events”. As requested by reviewer #1, the updated figure also distinguishes between clonal and sub-clonal shared mutations.

15. The relationship between ITH and the size of FL is intriguing but could the authors investigate this apparent link further?

Response:

We thank the reviewer for encouraging us to further investigate/discuss this association. As outlined in response to point #8 of reviewer #1, large FLs, especially in patients with >60% differences, did not share a specific mutation or CNA. However, they showed heterogeneity for at least one of the recently identified strong drivers of myeloma progression. Three of these 5 patients showed an *unshared* inactivation of the tumor suppressor *TP53* or *RB*. In addition, we found shared-diff mutations for the MAPK genes *NRAS* or *KRAS* in Patient #2 and #3. To illustrate this lack of a common “driver” and heterogeneity in progression events, we have updated Figure 5. According to our interpretation, advanced clones grow in FLs. Multiple “progression” events are likely required to become an advance clone and passenger mutations may accumulate at the same time. This could be an explanation for the association. We have updated our discussion accordingly (See response to 10&11)

Response to Reviewer #3

Rasche et al performed a spatial genomics analysis on a cohort of MM patients showing a significant heterogeneity across focal lesions, highlighting the importance of analyzing multi-region in order to better understand the intra-patient heterogeneity. The manuscript addresses a very relevant subject in MM biology that could potentially lead to an improvement in the risk stratification and personalized therapies. The manuscript is well written, the figures are clear, the experiments are done using cutting edge technology and the conclusions are supported by the data. I think it is a very relevant study that advances in the field of MM biology.

I have some few comments that need to be addressed in order to improve the clarity of the manuscript.

1. The investigators showed 2 cases where a hyperdiploid karyotype in differentially found across sites. The same applies to -13q. Both, -13q and hyperdiploid MM (H-MM) are very early and stable event in MM pathogenesis and, as long as I know, there are no clear reports showing a switch from H-MM to NH-MM or vice versa. It would be interesting that the authors elaborate a little more in the similarities and differences between those biopsies. Unfortunately, the way the data is presented precludes for performing a complete comparison. Reading lines 112-120 it is not clear what is the level of similarity of the karyotypes between sites either. For example: Is it possible that the 4th lumbar vertebra of case #1 shared trisomies in odd chromosomes with the iliac crest, but subsequently loss another chromosomes leading to the change from H-MM to NH-MM? The authors should show in suppl table 3, not only the non-shared abnormalities but also the shared abnormalities between sites for better understanding of the evolutionary tree.

Response:

First of all, we would like to thank the reviewer for the positive feedback!

We agree with the reviewer that the two cases with a discordant ploidy status needs to be presented in more detail. Thus, we have added Supplemental Figure 1 which illustrates the chromosomal profiles of the two patients and referred to it in the results paragraph.

Furthermore, we have included all *shared* and *unshared* abnormalities (>1000 events) in Supplemental Table 3.

Supplemental Figure 1: Chromosomal profiles. The plot illustrates the chromosomal profiles for paired samples of patient #1 and #5 who presented with discordance regarding the ploidy status.

- Is it possible that some of the differences between sites could be explained by the ratio of tumor/normal PCs collected in the guided versus non-guided aspirates? A priori, guided aspirates should be richer in tumor cells. On the other hand, it could be possible that the non-guided aspirates are diluted with normal PCs, which would affect the detection of abnormalities. That would be especially affecting the copy-number changes detections, considering the relatively low sensitivity offered by copy-number arrays. How did the authors measure the tumor purity in each sample? Have the authors validated the lack of CNAs by an independent approach, such as FISH?

Response:

A) We agree with the reviewer that CD138-enriched tumor samples can be diluted with normal (plasma) cells and that this “contamination” might impact the analysis. We used Sequenza to estimate the proportion of tumor cells based on whole exome sequencing data. Indeed, we detected a slight difference in the purity level between iliac crest (median 89%) and focal lesion (median 95%) samples. However, we neither detected a significant association between the purity and the number of detectable mutations nor the purity and the proportion of unshared mutations (linear model). The same holds true for CNAs.

B) We agree with the reviewer that lower purities may especially impact CNA calls. To account for that and the lower sensitivity of CNA analyses, we included only CNAs into the analysis that were detected as major aberrations in at least one of the paired samples and manually investigated copy number profiles of each sample.

C) Basically, for the majority of patients we already used two independent assays to investigate copy number profiles (whole exome sequencing and arrays). However, we agree that FISH is a quantitative approach that allows to investigate CNAs with a higher sensitivity. So far, FISH has been performed for random aspirates of newly diagnosed patients only in our center. FISH data for the regions 1p13, 1q21, and 17p13 were available for 30 patients (please see table below). We found 89 concordant results. We found one discordant result for 1q21. According to array data patient #38 was negative for gain(1q21) in the paired samples. Using FISH we found a sub-clone (proportion of ~20%) in the random aspirate by FISH. However, in 2 other cases with a sub-clonal gain of 1q21 in random aspirates according to array data, we also detected sub-clonal gains with a proportion of ~20% using FISH. Thus, we estimate that our threshold for CNA detection is ~20%. We have added this estimate to the results paragraph “Heterogeneity at the chromosomal level” and have also added a paragraph to “Methods” with a discussion of the technical limitations of our study.

1p12	Array: deletion	Array: w/o deletion
FISH: deletion	5	0
FISH: w/o deletion	0	25
1q21	Array: gain	Array: w/o gain
FISH: gain	10	1*
FISH: w/o gain	0	19
17p13	Array: deletion	Array: w/o deletion
FISH: deletion	3	0
FISH: w/o deletion	0	27
* sub-clone (frequency ~20%)		

Heterogeneity at the chromosomal level

We used high resolution SNP arrays and WES data to call copy number aberrations (CNAs). To account for the sensitivity of this approach for detection of sub-clones (threshold: ~20%) we only included CNAs that were clonal in at least one the paired samples. Using this strategy ...)

Limitations

The threshold for detection of *unshared* mutations and CNAs was approximately 1.5% and 20%, respectively. Thus, a *shared-diff* variant with a lower frequency than these cut-offs would be “misclassified” as *unshared*. To account for differences in the purity of samples (Supplemental Table 2) and the coverage of WES, we used stringent cut-offs (see above). As a result, our study rather underestimates spatial heterogeneity. Moreover, the limited set of samples per patient investigated in this study, potentially also led to an underestimation of heterogeneity.

3. The significant reduction of differences in the pre-treated samples compared with the newly diagnosed is an interesting observation reinforcing the evolutionary dynamics and clonal evolution of MM. The authors should discuss those observations in more detail.

Response:

A) We admit that, at first glance, Figure 2a and 2d give the impression that heterogeneity in space is reduced in treated patients. However, counting all events affecting driver genes in Figure 2d, 16/42 (38%) newly diagnosed and 4/11 (36%) treated patients showed heterogeneity in space. We also performed a Mann-Whitney Wilcoxon test and could not detect a difference in the level of spatial heterogeneity between these two sets of patients.

B) We agree with the reviewer that a discussion of these results was neglected in the first version of the manuscript. Thus, we added the following paragraph to the discussion section:

“This may also allow for a better interpretation of the results seen in treated patients. Regionally dominant clones after treatment could indicate multiple independent resistant clones. Alternatively, a minor sub-clone prior to therapy could be selected for. Further regional evolution of this clone could also lead to the type of spatial heterogeneity seen in treated patients. The latter is supported by the observation that 8 of 11 treated patients showed shared mutations in MAPK genes. Whether the mutagenic effect of chemotherapy has an impact on genomic spatial heterogeneity is still elusive. However, we could not detect a signature with the classic features of alkylating agents or cisplatin exposure in our recent study of patients relapsing after chemotherapy.”

4. Missing a thoughtful discussion of the lower heterogeneity in relapsed disease

Response:

Please see response to point #3.

5. Are the 4 cases showed in figure 4 the only cases with multiple FLs analyzed or additional cases were analyzed? Please clarify.

Response:

Aim of this analysis was to illustrate phylogenetic trees for patients with spatial heterogeneity and multiple samples. Data for multiple FLs were available for 6 patients and we reported on 4 of them. The two remaining cases presented with small (<2.5 cm) FLs only and subsequently with a low level of spatial heterogeneity. We modified the sentence in the manuscript to:

“To address whether the genomic profile of a single FL is representative of other FLs in the same patient, we investigated the phylogenetic relationship between clones at different sites in 4 patients with large FLs (>2.5 cm).”

6. Figure 4 is very nice. The idea of showing color variations for the different subclones makes sense but it makes a little more difficult the visualization of the medical images (especially fig 4A). I suggest changing the color codes of the subclones showed in fig 4A, C, and D.

Response:

We would like to thank the reviewer for this very positive feedback! We have changed the color code of Fig 4 a,c, and d. To further facilitate assignment of clones to anatomical positions, we included the caption “anatomical position” to Fig 4A and the caption “R” in the upper corner of the radiological images to indicate the right site of the body.

7. Supplementary tables 1, 2, 4 and 6 have format issues in the pdf version. Please correct

Response:

We thank the reviewer for pointing to this issue. We have corrected the tables.

REFERENCES

1. Raab, M.S. *et al.* Spatially divergent clonal evolution in multiple myeloma: overcoming resistance to BRAF inhibition. *Blood* **127**, 2155-7 (2016).
2. Lopez-Anglada, L. *et al.* P53 deletion may drive the clinical evolution and treatment response in multiple myeloma. *Eur J Haematol* **84**, 359-61 (2010).
3. Williams, M.J., Werner, B., Barnes, C.P., Graham, T.A. & Sottoriva, A. Identification of neutral tumor evolution across cancer types. *Nat Genet* **48**, 238-44 (2016).
4. Gerlinger, M. *et al.* Intratumor heterogeneity and branched evolution revealed by multiregion sequencing. *N Engl J Med* **366**, 883-92 (2012).
5. Hao, J.J. *et al.* Spatial intratumoral heterogeneity and temporal clonal evolution in esophageal squamous cell carcinoma. *Nat Genet* (2016).
6. Ling, S. *et al.* Extremely high genetic diversity in a single tumor points to prevalence of non-Darwinian cell evolution. *Proc Natl Acad Sci U S A* **112**, E6496-505 (2015).
7. de Bruin, E.C. *et al.* Spatial and temporal diversity in genomic instability processes defines lung cancer evolution. *Science* **346**, 251-6 (2014).
8. Zhang, J. *et al.* Intratumor heterogeneity in localized lung adenocarcinomas delineated by multiregion sequencing. *Science* **346**, 256-9 (2014).
9. Gibson, W.J. *et al.* The genomic landscape and evolution of endometrial carcinoma progression and abdominopelvic metastasis. *Nat Genet* **48**, 848-55 (2016).
10. Avva, R., Vanhemert, R.L., Barlogie, B., Munshi, N. & Angtuaco, E.J. CT-guided biopsy of focal lesions in patients with multiple myeloma may reveal new and more aggressive cytogenetic abnormalities. *AJNR Am J Neuroradiol* **22**, 781-5 (2001).
11. Lohr, J.G. *et al.* Genetic interrogation of circulating multiple myeloma cells at single-cell resolution. *Sci Transl Med* **8**, 363ra147 (2016).
12. Mithraprabhu, S. *et al.* Circulating tumour DNA analysis demonstrates spatial mutational heterogeneity that coincides with disease relapse in myeloma. *Leukemia* (2017).
13. Morgan, G.J., Walker, B.A. & Davies, F.E. The genetic architecture of multiple myeloma. *Nat Rev Cancer* **12**, 335-48 (2012).
14. Manier, S. *et al.* Genomic complexity of multiple myeloma and its clinical implications. *Nat Rev Clin Oncol* (2016).
15. Robertson-Tessi, M. & Anderson, A.R. Big Bang and context-driven collapse. *Nat Genet* **47**, 196-7 (2015).
16. Gerlinger, M. *et al.* Genomic architecture and evolution of clear cell renal cell carcinomas defined by multiregion sequencing. *Nat Genet* **46**, 225-33 (2014).
17. Lohr, J.G. *et al.* Widespread genetic heterogeneity in multiple myeloma: implications for targeted therapy. *Cancer Cell* **25**, 91-101 (2014).
18. Bolli, N. *et al.* Heterogeneity of genomic evolution and mutational profiles in multiple myeloma. *Nat Commun* **5**, 2997 (2014).
19. Walker, B.A. *et al.* Mutational Spectrum, Copy Number Changes, and Outcome: Results of a Sequencing Study of Patients With Newly Diagnosed Myeloma. *J Clin Oncol* **33**, 3911-20 (2015).
20. Weinhold, N. *et al.* Clonal selection and double-hit events involving tumor suppressor genes underlie relapse in myeloma. *Blood* **128**, 1735-44 (2016).
21. Johnson, S.K. *et al.* The use of molecular-based risk stratification and pharmacogenomics for outcome prediction and personalized therapeutic management of multiple myeloma. *Int J Hematol* **94**, 321-33 (2011).
22. Meissner, T. *et al.* Profound impact of sample processing delay on gene expression of multiple myeloma plasma cells. *BMC Med Genomics* **8**, 85 (2015).
23. Bolli, N. *et al.* A DNA target-enrichment approach to detect mutations, copy number changes and immunoglobulin translocations in multiple myeloma. *Blood Cancer J* **6**, e467 (2016).

Reviewers' Comments:

Reviewer #1:

Remarks to the Author:

This is the revised submission by Chavan et al reporting on Spatial Genomic Heterogeneity in Multiple Myeloma identified by Multi-Region Sequencing. The manuscript has been revised; however most of the responses are more of theoretical discussion rather than providing new analysis and data. For example

1) Response to the question "the interpretations, two phase model of progression and clonal sweeps, are not supported by the data. In this descriptive manuscript there is no confirmation of any of these findings." The response is still all assumptions based on author observations without providing the detailed data.

2) In response to "clonality and subclonality of observed mutations have not been quantitated and in absence of that it will be difficult to conclude regarding clonal drift as well as sweep"; the authors claim that now they do incorporate subclonal changes. Figure 2C now shows that subclonal mutations or shared-diff mutations contributes to majority if not all the cases. But authors still fail to show that how big the differences between multiple locations. It is likely that all subclonal mutations would be just at the detection level and one would miss such low level subclones at other locations simply because of technical reasons. Authors should provide further evidence on mutational differences between locations.

3) A deeper sequencing and/or PCR confirmation, was requested to confirm the results. This is not provided. Authors keep using different explanations for VAF and clonality. They claim that they use 5% VAF (without correction VAF do not represent the clonality and even with 100% purity and diploidy 5% VAF equals to 10% clonality) to call mutations and to avoid the overestimation they increase cancer cell fraction to 0.2 which is 10% VAF with 100% purity and diploidy. Later they claim they increase read depth to 50 but with 50X coverage one would barely call < 10% mutations and finally check 1.5% VAF for unshared mutations.

4) The depth of sequencing (142X) is still very low for the conclusions. The response that " Our coverage is within the range of reported coverages", is not acceptable. The earlier studies were meant for reporting the mutational spectrum, while the current study tries to report spatial distribution of clones. A greater depth at least in subset of samples is essential to comment on lower level presence of clones at different areas to judge clonal distribution.

5) The Aim of this study is different than other publications in the field. Study specially focuses on subclonal changes and to make findings accurate authors should have to have deeper sequencing than others. They claim that 142X is the average depth but they call mutations in the region where they have 50X coverage and more!

5) R1 Q6: Authors have not answered the concordance question.

6) R1 Q7: The answer to the question is missing.

7) R1 Q9: What is the power in the study?

8) R1 Q12: It seems that GEP70 on multiple sites have failed. This is contrary to their claim of uniform application of GEP70.

Reviewer #2:

Remarks to the Author:

The authors have thoroughly addressed my comments and this paper should be accepted for publication

Reviewer #3:
None

Response to Reviewer #1

1) Response to the question “the interpretations, two phase model of progression and clonal sweeps, are not supported by the data. In this descriptive manuscript there is no confirmation of any of these findings. ”The response is still all assumptions based on author observations without providing the detailed data.”

Response

The reviewer is correct that experimental studies investigating evolutionary processes in multiple myeloma and providing direct evidence for the mechanisms underlying these processes are of high interest. Consequently, we also think that discussing our results in the context of an illustrative model, which integrates our observations of regional evolution into the current concept of clonal sweeps in multiple myeloma, can facilitate the understanding of our translational data, which is derived from the study of primary patient samples. Actually, the type of Darwinian model we discuss simply takes into account the impact of spatial constraints on evolutionary processes; a concept previously used to explain both the evolution of solid cancers and bacteria. We agree with the reviewer that an experimental proof of this novel concept in multiple myeloma will be an important step but this would go well beyond the scope of this paper which aimed to describe spatial heterogeneity.

In addition, to the best of our knowledge, direct evidence for the existence of sweeps at the early stages of disease and subsequent regional evolution can (currently) not be provided either by *in vitro* or *in vivo* studies for the following reasons:

- 1) Longitudinal data for MGUS progression is limited due to the low rate of progression and availability of samples.
- 2) The exact location and time point of MGUS “initiation” is not known.
- 3) A prospective analysis of focal lesion evolution starting at the “initiation” phase is not possible, since the sites of future focal lesions cannot be predicted.
- 4) In our opinion, given the nature of model systems for myeloma, *in-vitro* or animal experiments would not be representative of disease evolution in patients. There is no myeloma cell line that is representative of early disease stages and as a consequence cannot be used to study the evolutionary processes described in our paper. Established animal models are highly artificial and it is not possible to engraft primary human tumor cells from early disease stages.

Collectively, we think that the interpretation of data obtained from patient derived tumor cells is currently the best way to gain insight into the evolutionary processes active in MM. As discussed in our manuscript, we also think that a combination of spatial and longitudinal analyses could significantly enhance our understanding of these processes.

Although our data strongly supports non-neutral evolution in myeloma, we believe that at the same time it enhances a “simple” Darwinian type evolution model characterized solely by clonal competition and subsequent sweeps. The data presented supports the existence of

sweeps, at least at early disease stages (shared progression events and complex trunks being less frequently seen in MGUS; shared-diff mutations). We interpret the trend to a lower proportion of shared-diff mutations in advanced (ISS III) disease and regional differences in the context of complex/advanced aberrations as evidence for a decreased capacity for full clonal “sweeps” at later stages of the evolutionary process. In our opinion, the co-existence of ultra-high risk and low risk clones in newly diagnosed patients such as patient #1 also strongly supports our interpretation that clonal sweeps are limited at later disease stages, where spatial constraints would be greater.

We clearly demonstrate regional differences with our data. The best example of this is patient #8 (Fig. 4b) who presented with 4 distinct clones at 4 different sites but also with the shared progression events t(MYC) and amp(1q21). In our opinion, clonal sweeps followed by regional evolution is an adequate interpretation of these observations. This case is presented in detail in the manuscript.

To further address the reviewer’s critique and to clarify that we have discussed our observations in the framework of an existing evolutionary concept, we have modified our discussion. Specifically, we have edited or added the following sentences:

- In Figure 5 we illustrate these considerations in the context of a non-neutral progression model with two main phases that may explain our observations in MM patients with FLs: the *first phase* is characterized by selective sweeps of more advanced “fitter” clones replacing previously dominant sub-clones. This idea is supported by our observation that progression events such as gain of 1q are frequently ubiquitously distributed in newly diagnosed MM patients; an aberration less frequently seen at the monoclonal gammopathy of undetermined significance (MGUS) stage^{1,2}. It is also supported by recently published evidence for clonal expansion and an increase in the number of mutations during the progression from MGUS to MM^{1,3}.
- The concept of spatial constraints limiting the capacity for selective sweeps was recently introduced for bacteria and solid tumors⁴⁻⁶, and it is very appealing to explain our observations in MM, where bone marrow survival niches are limited.
- Of note, while derived from patients with FLs, “regional progression” could also occur in cases without FLs as we see cases lacking FLs, that show considerable heterogeneity in space and as the number of sites we investigated was limited.
- According to our interpretation, highly advanced clones growing as FLs should contain strong and more numerous driver events.
- Importantly, our approach represents a snapshot of ongoing evolution and could be considerably enhanced by using a combination of longitudinal and spatial investigations, starting at premalignant stages of the disease, to give a more complete picture of the complex evolutionary processes.
- However, other mechanisms could result in the development of FLs including local differences in the tumor microenvironment selecting for clones with distinct genomic aberrations⁷. Thus, in the future it will be important to investigate the interactions between tumor cells and their microenvironment in FLs and compare these to FL-free sites.

2) In response to “clonality and subclonality of observed mutations have not been quantitated and in absence of that it will be difficult to conclude regarding clonal drift as well as sweep”; the authors

claim that now they do incorporate subclonal changes. Figure 2C now shows that subclonal mutations or shared-diff mutations contributes to majority if not all the cases. But authors still fail to show that how big the differences between multiple locations. It is likely that all subclonal mutations would be just at the detection level and one would miss such low level subclones at other locations simply because of technical reasons. Authors should provide further evidence on mutational differences between locations.

Response

We agree with the reviewer that the exact level of differences between paired samples, especially for shared-diff mutations, cannot be seen in Figure 2C, and we apologize that we did not fully address this in our response. To better address the request we have now plotted the cancer clonal fraction (CCF) for the “positive” sample and the read depth of the “negative” sample at the respective mutation sites for both clonal and subclonal mutations (Supplemental Figure 1B & C). We also illustrate the CCF difference for shared-diff mutations (Supplemental Figure 1D). We show that 18 of 42 newly diagnosed patients present with at least 1 clonal unshared mutation, indicating that in these patients unique main clones dominate at different sites (C). Furthermore, 15 of these 18 patients have a coverage of >142 (the study-wide depth average) with a maximum of 410x at the corresponding negative sites (B), illustrating that our observation of site-unique clones is not solely the result of technical issues. This point is also supported by the new deep-WES data we have generated and utilized in the 2nd revised version and the results of targeted sequencing which had already been presented in the 1st revised version of the manuscript. We also illustrate this data further in our response to question 4.

In summary, none of the unshared mutations was just at the detection level in the “positive” sample. Thus, the inability to detect mutations with a good depth of sequencing at other sites strongly supports regional differences.

3) A deeper sequencing and/or PCR confirmation, was requested to confirm the results. This is not provided. Authors keep using different explanations for VAF and clonality. They claim that they use 5% VAF (without correction VAF do not represent the clonality and even with 100% purity and diploidy 5% VAF equals to 10% clonality) to call mutations and to avoid the overestimation they increase cancer cell fraction to 0.2 which is 10% VAF with 100% purity and diploidy. Later they claim they increase read depth to 50 but with 50X coverage one would barely call < 10% mutations and finally check 1.5% VAF for unshared mutations.

Response

We apologize that this was not clearly presented in the 1st response letter, as targeted sequencing data was provided for 31 patients in the 1st revised version of the manuscript. Please see response to question 4 (“*A greater depth at least in subset of samples is essential!*”) for a detailed response to the confirmation request and a description of the new deep-WES data.

We agree with the reviewer that the use of VAF and CCF at different steps of our filtering strategy could be confusing for the reader and yet was based on established/standard procedures. We explain these in more detail for clarity:

- 1) We used the `pfILTER.pl` script and filtered for low frequency variants to reduce the number of false-positive mutation calls. Since the background “noise” in sequencing

experiments is not related to clonality and the script uses read numbers as input, this filtering step is based on the number of “alternative” and total reads. Here we used a threshold of 5% VAF; the default for this script.

- 2) The next filtering step was based on the proportion of subclones. Thus, we continued to work with CCF values which are corrected for purity and copy number. To avoid the issue that “*subclonal mutations could just be at the detection level and would be missed (in the paired sample) simply because of technical reasons*” we used a rather conservative threshold of 20% CCF for the “positive” sample and at least 50 reads in the paired “negative” sample. We increased both values, because, for example, 2 alternative reads at a total read number of 20 would also correspond to a CCF of 20% (diploid region) but would indeed be at the detection level. In contrast, with 50 total reads and a threshold of 2 alternative reads, the detection threshold would be below 20% CCF (8% at 50 reads). This is not an unreasonable approach as a threshold of 50x coverage was recently used by Sottoriva *et al.* in order to “avoid overcalling ITH as a result of false negatives due to low coverage.”⁶

For convenience, we stated an average VAF frequency in the results paragraph, because 2 reads/total number of reads could correspond to different CCFs, depending on the sample purity and the copy number at the respective site(s). However, we agree with the reviewer that simply stating a VAF average value could be misleading. Therefore, in order to increase clarity, we have:

- (A) removed the VAF detection threshold from the results paragraph and referred to the methods paragraph,
- (B) added the VAF threshold range to the method section,
- (C) indicated the corresponding average CCF values, and
- (D) also referred to Supplementary Figure 1, which shows the total number of reads at the “negative” site for each clonal and subclonal “unshared” mutation.

4) The depth of sequencing (142X) is still very low for the conclusions. The response that “Our coverage is within the range of reported coverages”, is not acceptable. The earlier studies were meant for reporting the mutational spectrum, while the current study tries to report spatial distribution of clones. A greater depth at least in subset of samples is essential to comment on lower level presence of clones at different areas to judge clonal distribution.

Response

We have taken the reviewers point on board and as requested we have performed deep whole exome sequencing on a HiSeq2500 for a subset of samples (n=11, 4 patients) in order to exclude that apparent regional differences were due to a lack of sequencing depth. Using this strategy the average depth was ~500x at the sites of unshared non-silent mutations. To the best of our knowledge, this is the highest coverage provided for mutations in myeloma investigated by WES. Of note, these values were calculated after removal of sequencing read duplicates; a filtering step that is not possible using a standard amplicon sequencing strategy.

This additional analysis showed that for 85 of the 90 unshared non-silent mutations in these patients the respective mutation could not be detected at the paired “negative” site, even using an ultra-conservative threshold of 2 alternative reads. For the 5 remaining mutations we found between 2 and 4 alternative reads at the paired site, highly suspicious for false-positive

results (noise). Of note, in patient #8 unshared clonal mutations were not detectable in the paired samples with a depth of up to >1000x (please also see the Figure below and Supplemental Table 7).

To further address the issue of sequencing depth, we present targeted sequencing data for 31 patients using an FDA approved test (this data was included in the 1st revised version of the manuscript in response to question 4). Among others, this test confirmed the absence of a (clonal) *BRAF* mutation in a paired “negative” sample at a depth of 650x in patient #1.

In summary, we have produced additional data to address the reviewers concern. The results give a high level of confidence that regionally restricted aberrations do indeed exist in myeloma and that our interpretations are strongly supported by data.

Supplemental Figure 8: Confirmation of unshared mutations at higher depth. The figure shows the results of deep WES sequencing for patient #8. The unshared non-silent mutations in the driver genes *BRAF*, *KRAS* and *STAT3* (*Asn553Lys, **Asp661Tyr) and the other genes *CYP27B1* and *FGF12* were selected as representative examples. The cancer clonal fraction (CCF) at the positive sites and the sequencing depth at the negative sites are shown in brackets. Please also see Supplemental Table 7 for a complete overview of deep WES confirmation of non-silent unshared mutations in patient #3, #7, #8, and #19.

5) The Aim of this study is different than other publications in the field. Study specially focuses on subclonal changes and to make findings accurate authors should have to have deeper

sequencing than others. They claim that 142X is the average depth but they call mutations in the region where they have 50X coverage and more!

Response

We thank the reviewer for giving us the possibility to clarify this point. The study-wide average depth at sites of unshared mutations was 142 with a range of 50-540x. We used a threshold of 50x to avoid an overestimation of heterogeneity but also to avoid the loss of mutations in regions that were also deleted, such as *TP53* mutations (a deleted region would present with 50% of the average coverage). As also outlined above in response to point 3, this strategy was recently used by Sottoriva *et al.*⁶ At a 50x depth, we fully appreciate that the detection threshold would be significantly lower than the study-wide average detection threshold of ~1.5 VAF. To account for this, we have changed the manuscript and added a VAF detection threshold range to the methods paragraph. Despite the average depth being relatively low, a subset of patients had clonal unshared mutations with read depths up to 400x at the negative site, with the same holding true for unshared subclonal mutations (up to 540x), as illustrated in Supplemental Figure 1B-C.

In summary, we are confident that our conclusion that distinct major clones may dominate at different sites, is supported by read depths that are of sufficient sensitivity to exclude the presence of clones. To reassure the reviewer we also sequenced a subset of cases to a higher depth which supports this conclusion. Please also see response to question 4 (“*A greater depth at least in subset of samples is essential*”).

6) R1 Q6: Authors have not answered the concordance question.

Response

We apologize for not adequately addressing the question: “*What is the concordance between FNAS and RNAS for cytogenetic data?*” and we attempt to do this more fully below.

As reviewer #3 had specifically requested an update of Supplemental Table 3, we described all unshared and shared (large scale) copy number aberrations per patient and sample in this table in the 1st revised version of the manuscript.

To further address this concordance question, we have now plotted the concordance rate [%] in Supplemental Figure 1A. On the one hand, up to 76% of CNAs were not shared between paired FNAS and RNAS samples. On the other hand, in 25 patients 100% CNAs were shared. We also point readers to Supplemental Table 3 for further details: in patients with a low number of CNAs even a single unshared CNAs might have a high impact on the concordance level (e.g. patient #14 with 1 shared and 1 unshared event).

7) R1 Q7: The answer to the question is missing.

Response

We apologize for this oversight and address the question: “*The majority of clonal differences are presented as large size cytogenetic changes. Does this mean there are no significant mutational differences between these various regions?*” here. All variants in Figure 4 are clonal shared or unshared events. Thus, subclonal mutations and CNAs are not shown in this figure. However, despite this focus on clonal events, the majority of differences were seen on the mutational level.

For example, for patient #12 we found 17 branch-specific (clonal) mutations but “only” 5 unshared copy number aberrations. For all 4 patients presented in Figure 4 the differences were more significant at the mutational level. This higher level of heterogeneity at the mutational level is presented in Figure 2 for the complete set of patients and is stated in the results paragraph: “The analysis of mutational profiles showed that genomic heterogeneity in space was more pronounced than was seen at the CNA level (Fig. 2c).” [page 6, line 133-134]

8) R1 Q9: What is the power in the study?

Response

To address this question which refers to the previous Q9: “*The differences between size and unshared mutations may reflect variation due to sampling size rather than true biological differences.*” we performed a power analysis using $n=42$. For a Spearman’s correlation we had 80% power to detect an effect (correlation coefficient) of 0.42 at the significance level of 0.05.

9) R1 Q12: It seems that GEP70 on multiple sites have failed. This is contrary to their claim of uniform application of GEP70.

Response

For each of the 263 patients, paired samples were available and the GEP70 risk status (of paired samples) was determined for all of them. All samples were processed using the same published standard procedures: CD138-positive selection of aspirates, gene expression profiling analysis using Affymetrix U133 Version 2.0 chips, and calculation of the GEP70. Of note, our genomic analysis of a subset of patients with GEP70 discrepancies rather supports the existence of spatial heterogeneity for high risk/advanced clones, since we found unshared progression (poor prognostic) events in these patients, e.g. gain or amp 1q21, del(17p), t(MYC) and/or bi-allelic deletions of tumor suppressor genes in the newly diagnosed patients #1, #3, #12, and #24, and the treated patients #43, #44, and #48. Furthermore, patients #5 and #18 presented with heterogeneity impacting the ploidy status and/or myeloma driver genes.

In conclusion, we are confident that differences in the risk score and the result of the survival analysis reflect true biological differences and are not due to technical issues.

REFERENCES

1. Lopez-Corral, L. *et al.* The progression from MGUS to smoldering myeloma and eventually to multiple myeloma involves a clonal expansion of genetically abnormal plasma cells. *Clin Cancer Res* **17**, 1692-700 (2011).
2. Rajan, A.M. & Rajkumar, S.V. Interpretation of cytogenetic results in multiple myeloma for clinical practice. *Blood Cancer J* **5**, e365 (2015).
3. Walker, B.A. *et al.* Intraclonal heterogeneity is a critical early event in the development of myeloma and precedes the development of clinical symptoms. *Leukemia* **28**, 384-90 (2014).
4. Korolev, K.S. *et al.* Selective sweeps in growing microbial colonies. *Phys Biol* **9**, 026008 (2012).
5. Baym, M. *et al.* Spatiotemporal microbial evolution on antibiotic landscapes. *Science* **353**, 1147-51 (2016).

6. Sottoriva, A. *et al.* A Big Bang model of human colorectal tumor growth. *Nat Genet* **47**, 209-16 (2015).
7. Greaves, M. & Maley, C.C. Clonal evolution in cancer. *Nature* **481**, 306-13 (2012).

Reviewers' Comments:

Reviewer #1:

Remarks to the Author:

The authors have addressed my comments